# Measuring the Transit-Oriented Development (TOD) Levels of Pakistani Megacities for TOD Application: A Case Study of Lahore

**Ayesha Anwar** [1,2,3,*] 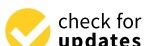, **Hong Leng** [1,2,*], **Humayun Ashraf** [4] and **Alina Haider** [3]

1    School of Architecture and Design, Harbin Institute of Technology, Harbin 150001, China
2    Key Laboratory of National Territory Spatial Planning and Ecological Restoration in Cold Regions, Ministry of Natural Resources, Harbin 150001, China
3    Department of Architecture, University of Gujrat, Gujrat 50700, Pakistan; alina.haider@uog.edu.pk
4    Department of Geography, University of Gujrat, Gujrat 50700, Pakistan; humayun.22@uog.edu.pk
*    Correspondence: ayesha@hit.edu.cn (A.A.); hitlaura@126.com (H.L.)

**Abstract:** The urbanization challenges in the megacities of Pakistan necessitate the implementation of comprehensive sustainable development practices to effectively address contemporary urban issues. Transit-oriented development (TOD) is a globally accepted device in achieving sustainable urban development through transport and land use integration. Evaluating the levels of TOD built in present conditions is essential for productive TOD planning, as it enables the prioritization of development interferences. In this context, we utilized a methodology to evaluate the levels of TOD (TOD-ness) present near transportation nodes through a TOD index. It utilizes ArcGIS and spatial multi-criteria analysis (SMCA) to determine the extent of TOD-supporting qualities around a transit node and identifies areas for potential improvements in transit orientation. The methodology was executed in the megacity of Lahore, situated in Pakistan. A TOD index was computed for areas surrounding the 26 LRT and 27 BRT stations along two existing corridors. The findings suggest that the TOD concept is feasible for Pakistani megacities, and urban decision makers can utilize the TOD index results to facilitate urban- or regional-level planning, funding, and investment policies. Furthermore, these findings offer valuable insights into the transportation obstacles and potential opportunities in similar developing cities in South Asia.

**Keywords:** megacities; sustainable development; spatial multi-criteria analysis (SMCA); transit-oriented development (TOD); TOD index; TOD-ness

## 1. Introduction

New sustainable urban development trends emphasize dense, proximate communities with interconnected public transportation, aligning with transit-oriented development (TOD) principles [1]. TOD has globally demonstrated success in enhancing urban and transportation sustainability [2–6]. TOD planning promotes walking, cycling, and public transit over car usage by fostering mixed-use neighborhoods near transit hubs with medium to high densities and pedestrian-friendly environments. Its benefits include urban sustainability, reduced reliance on automobiles, well-designed communities, improved access to public transit, walking, and cycling, as well as decreased car usage, obesity, and related health issues.

TOD revolves around T (transit), O (oriented), and D (development), emphasizing well-orientation towards the transit system for optimal benefits. Without this orientation, transit-adjacent development (TAD) may arise, lacking interaction with transit. In this regard, effective regional planning and policy should address two key issues: identifying areas with high transit orientation but limited public transport access (Potential TOD) and recognizing areas with excellent transit access but poor orientation toward transit (Actual TOD) [7].

This paper focuses on calculating the Actual TOD index using SMCA to improve existing conditions. Additionally, it can aid governments and private investors in prioritizing future project investments. It is crucial to conduct a quantitative assessment of TOD projects around transit nodes before practical implementation. Evans and Pratt [8] and J.L. Renne [9] emphasized the need for comprehensive measurement tools like the TOD index to evaluate areas for their "TOD-ness". "TOD-ness" refers to how well existing conditions at TOD sites align with established TOD principles, often necessitating significant investment and attention, as supported by references [10–12].

Cities in developing nations face greater susceptibility compared to their developed counterparts. They must navigate various obstacles, including prioritizing short-term transportation needs over long-term sustainable urban development plans, disjointed institutional frameworks, challenges in cross-sector coordination, and regulatory constraints. Research on practical TOD implementation primarily originates from developed countries (e.g., [7,11,13–15]), with limited prior experience and evidence supporting its applicability in developing cities, which necessitate context-specific strategic action plans.

For these reasons, this study attempts to fill the gap through particular studies about TOD application in Pakistan, a developing country in South Asia, majorly suffering from the transport sector and consequent environmental and urban issues. According to the 2017 census, Pakistan has two megacities with over 10 million people. Karachi, in Sindh province, is the largest with around 15 million people, followed by Lahore in Punjab province with 11 million people. This research emphasizes the planning parameters that require special attention to alleviate TOD as a sustainable transport strategy for addressing urban issues in megacities, using Lahore as a pertinent case study.

Lahore city is in global spotlights as the "World's most polluted city" [16], battling smog for years. This poses health risks, damages the environment, and requires stricter emission regulations in various sectors. The transportation sector is the primary source of emissions (Figure 1) and poor air quality in Lahore, with the largest contribution coming from two-stroke vehicles, followed by cars, jeeps, and wagons [17]. The significant rise in registered vehicles in Lahore from 2011–2021 is due to a preference for private transportation over mass transit, as shown in Figure 2. In this regard, it is imperative to evaluate Lahore's readiness for implementing TOD, which could act as a remedy for the aforementioned challenges. Several authors [18–22] have emphasized the necessity of TOD in their studies on Lahore and its new Mater plan for 2050 also suggests it.

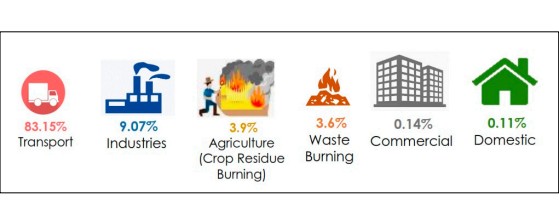

(**a**)

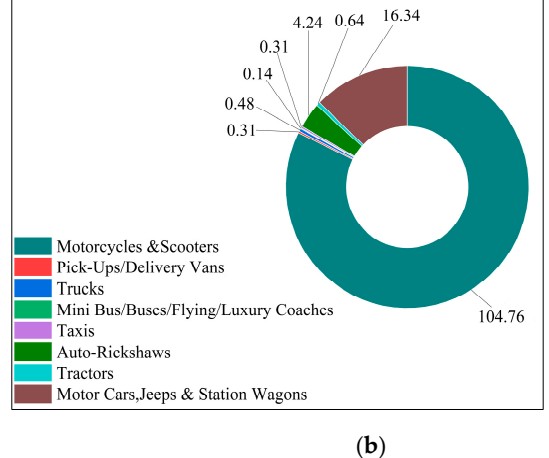

(**b**)

**Figure 1.** (**a**) Sectoral contribution of pollutants and (**b**) vehicular category-wise emissions. Source: Sectoral Emission Inventory of Lahore by Urban Unit.

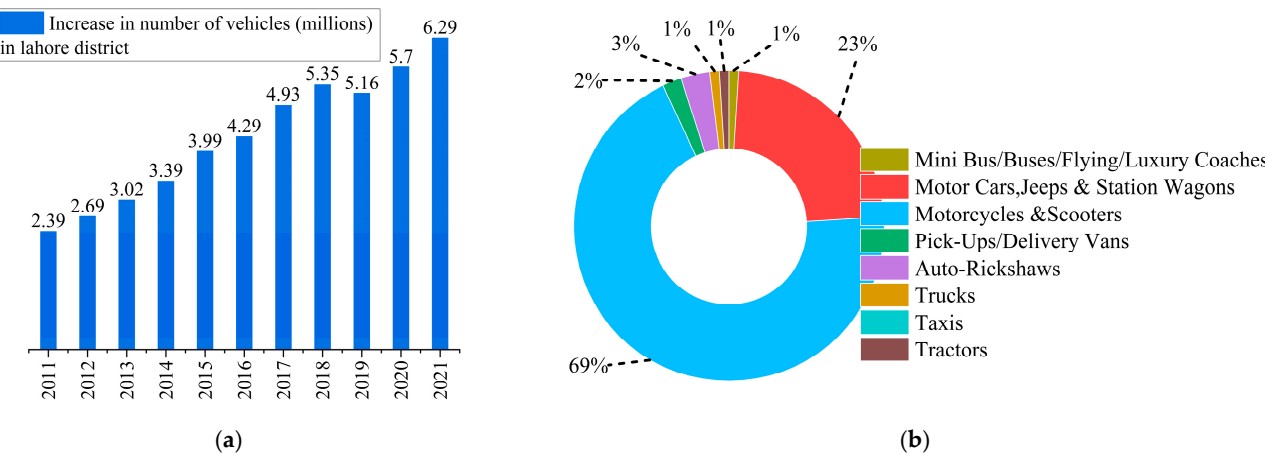

**Figure 2.** (**a**) Increase in no. of vehicles in Lahore, (**b**) vehicular category-wise increase. Source: Sectoral Emission Inventory of Lahore by Urban Unit.

To do so, we adopted the TOD index proposed by "Evans et al., 2007 and Singh et al., 2017" [8,11] to measure the TOD-ness of existing transit nodes within walking distance (500m buffer). Measuring the TOD using an index involves combining transit (T), orientation (O), and urban development (D) factors, which can vary in scale and nature, requiring multiple criteria analysis (MCA) for comprehensive assessment. Scholars often prefer spatial multi-criteria analysis (SMCA) for prioritizing intervention sites and understanding TOD levels at a regional scale. The study conducted by Teklemariam et al. [15] in Addis Ababa, Ethiopia is significant within the context of developing countries. However, it lacks key transport-related indicators crucial for determining TOD levels. While inspired by previous work, our study introduces a unique methodology for quantitative analysis and comparison of stations with diverse surrounding land uses. This methodology is empirically tested on a specific case study, signifying its applicability and linked qualities.

The methodology is applied to 27 BRT stations (Green line) and 26 LRT stations (Orange line) to identify areas requiring improved connectivity or orientation. The study sets the following objectives to achieve its main goal:

- Applying spatial and non-spatial analytical methods (ArcGIS and SMCA) to obtain the TOD index based on a set of indicators;
- Applying spatial statistical methods (sensitivity analysis) to analyze the obtained values of the TOD index;
- Guiding decision makers to recognize priorities for potential use.

The remaining paper is organized as follows: Section 2 provides a literature review of TOD assessment studies. Section 3 outlines our research design, methodology, and cri-teria selection for deriving the TOD index. Section 4 presents the case study for a better understanding of the study area. Sections 5–7 detail the TOD index results, analysis, and interpretation. Finally, Section 8 concludes by highlighting the policy implications of the research findings.

## 2. Literature Review

Combining TOD planning across different levels is vital for tackling transportation challenges. Assessing current and proposed TOD levels around transit stations helps policymakers compare node performance pre- and post-TOD implementation [23]. Various approaches exist for studying station areas, including spatial analytical, spatial statistical, and simulation-based methods. Evans et al. and Singh et al. [8,11] established a TOD index to calculate the current level of TOD of transit stations. Likewise, Higgis and Kanaroglou [24] worked on TOD typologies of 372 stations in the Toronto region using the latent class method, and Shirke et al. [25] assessed the TOD impact along Mumbai metro Line-1 using the discrete choice model.

A vast literature on TOD evaluation is based on quantitative and qualitative analysis, and different authors suggest several indicators to build on the empirical or theoretic basis or context. A commonly used model in the literature is the node place model, which focuses on station typologies where all stations are grouped into types [26–28] based on the ideal balance between node and transit indices. Nevertheless, its effectiveness is limited because different studies have different definitions of goals, leading to distinct sets of typologies and restricted applicability and transferability of developed TOD typologies. Furthermore, when working with clustering/typologies, a set of recommendations is formulated and implemented for all stations within each typology. It is important to recognize that although station areas may have similarities, they are not identical. Consequently, the effectiveness of improvement recommendations may vary, given the absence of a universally applicable solution.

Some authors used a mixed method approach (qualitative and quantitative) for TOD evaluation [29–33]. Nagari et al. [29] investigated the connectivity and accessibility of Kali, Besar Jakarta, Indonesia, while Mohamad et al. [30] developed a typology for Batu Pahat, Malaysia. Similarly, Millard [31] assessed the development results of TOD plans in San Francisco and Seattle while Liu et al. [32] studied the effects of TOD on urban sprawl in Tokyo. Maheshwari et al. [33] evaluated pre- and post-TOD levels around planned Metro corridors in Ahmedabad, India and Jones [34] discussed the impacts of TOD on gentrification in Vancouver. Liu et al. [35] enhanced the conventional 3D approach for TOD typologies by adding spatiotemporal activity dynamics at transit stations in New York City by applying a self-organizing map (SOM). Mirzahossein et al. [36] used the analytic hierarchy process (AHP) to propose an active transport network for Qazvin, Iran. Similarly, a study by Phani et al. [37] employed the enhanced fuzzy-analytical hierarchical process (EFAHP) framework to address a TOD-planning problem in Delhi, India.

Although the aforementioned papers have demonstrated significant contributions in the evaluation of station surroundings, we have chosen to focus on creating a TOD index, which quantifies the TOD characteristics of every transit station using a numerical value, also emphasized by [8,11], rather than utilizing typologies. An obtained single value can be beneficial for objective comparison at various levels; regional, corridor, and site. Furthermore, every transit node can have specified exclusive recommendations to enhance its TOD-ness. Table 1 depicts previous studies on TOD calculation and their respective limitations. Furthermore, in Section 3, we have detailed the criteria derived from relevant literature for the selection of indicators and the methodology employed to compute the TOD index around transit stations.

**Table 1.** Highlighting limitations of previous work from the literature review.

| Source | Case Study Area | Approach | Limitations |
|---|---|---|---|
| The Institute of Transportation and Development Policy (ITDP) 2017 [38] | Version3.0 released in 2017, following previous versions released in 2013 and 2014 | It aims to evaluate and score urban development plans and projects based on adherence to eight TOD design principles: walk, cycle, connect, transit, mix, densify, compact, and shift. | The scoring system in urban design has limitations because it is subjective and relies on primary data collection. This makes its pertinency to station areas broader and more challenging than for individual projects or developments. |
| M. Hamid et al., 2018 [14] | Tehran, Iran | The study uses spatial analyses and a hierarchical fuzzy inference system (HFIS) for indicator computation and aggregation | The research data are limited to the year 2005, potentially hindering the accuracy and relevance of the model outputs to present-day conditions. |
| Teklemariam et al., 2020 [15] | Addis Ababa, Ethiopia | TOD index calculation for 22 LRT stations using eight indicators and SMCA method | The research holds significance within the context of developing countries; however, it overlooks the crucial aspect of criteria weighting. |

**Table 1.** *Cont.*

| Source | Case Study Area | Approach | Limitations |
|---|---|---|---|
| M. Uddin et al., 2023 [39] | Dhaka, Bangladesh | This work presents the nodal TOD index through eight indicators and objective-weighted spatial multi-criteria analysis (OSMCA) | The research addresses a substantial gap in the literature; however, it lacks a thorough sensitivity analysis for diverse scenarios. |
| Sara et al., 2023 [40] | Alexandria City, Egypt | Measured potential TOD using the SMCA method | The study has limitations as it evaluates based on only seven indicators. More precise results could be obtained by including additional indicators. |

## 3. Research Design

TOD should be evaluated and designed from a network system viewpoint, considering that the transit nodes are components that enhance the network's performance. This study aims to adapt and apply an existing method to assess the appropriateness of various locations as TODs (Actual TOD levels) using an index. This involves demonstrating the measuring method's abilities in a new setting and then conducting empirical testing in a case study. The developed method is an extension of a framework frequently utilized by different scholars [7,8,11]. The authors aim to utilize this method in a case study of Lahore to assess TOD levels, employing a TOD index to pinpoint areas in need of enhancement, as shown in Figure 3.

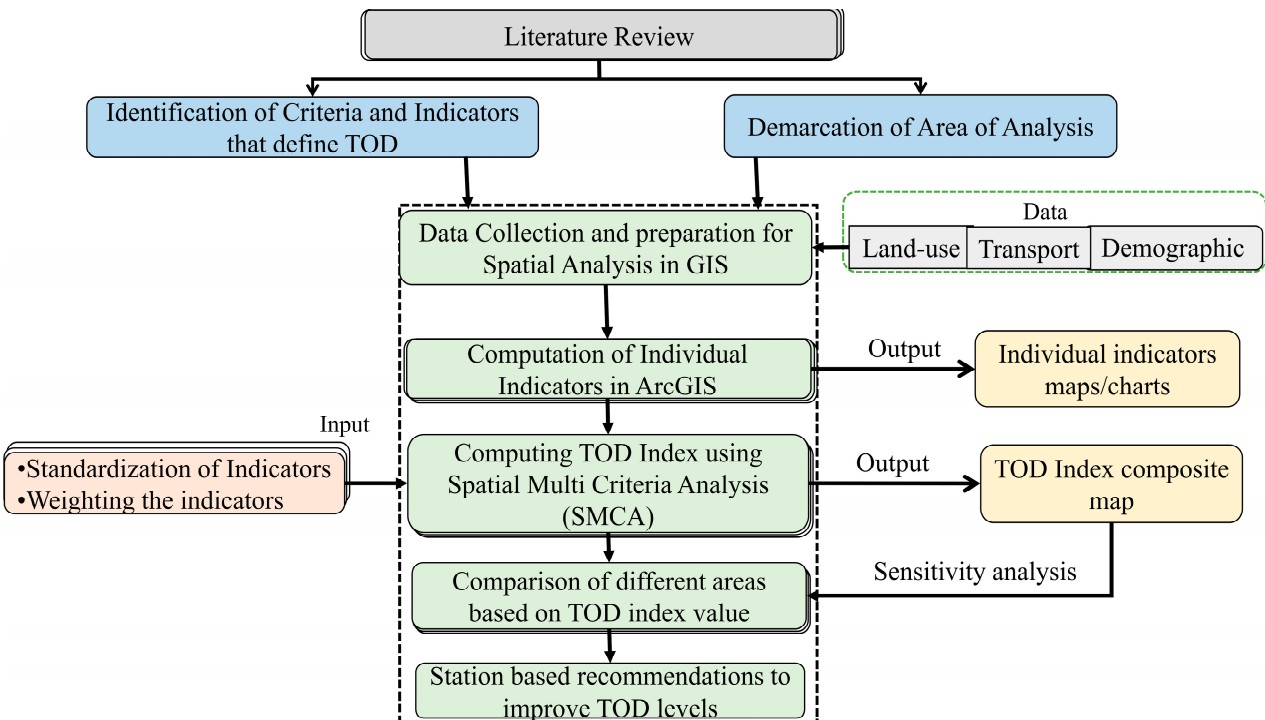

**Figure 3.** The overall research framework of the proposed study. Source: Author.

Our primary data collection involved on-site visits to gather performance measures, which serve as a benchmark for tracking future development. This method has limitations due to its time-consuming nature, as the accuracy of our work heavily relies on the collected data. The manual survey for primary data collection is challenging, and obtaining secondary data is difficult, especially when requiring location-specific information within a 500 m radius around the transit station. The case study analysis has yielded

valuable insights into key determinants of TOD and implementation, particularly around the station area.

### 3.1. Demarcation of Area of Analysis

An index should calculate TOD levels within walking distance of every transit node and it will measure characteristics of urban development and transit service, serving that area. So, before calculating the index, it is necessary to delineate the "area of analysis" or "TOD area" around every transit node, which is equivalent to walkable distance, which ranges from 400–800 m (5–10 min walking distance) per several references [38,41]. This range for a comfortable walk may vary from one region to another by geography or demographic studies.

In Lahore, it is known that people walk a distance of around 600 m to reach a public transport mode [42]. Distances greater than 500 m do not encourage people to meet their needs by walking and those from further away may take a paratransit service (rickshaw/Qingqi/feeder bus) to reach the main public transport mode. Some others may get a lift from a relative or friend to or from the public transport service.

Moreover, for transit lines in Lahore, the stations are spaced at approximately 1 km intervals along the developed sections of the alignment to reflect passengers' maximum walking distance of 500–600 m. Thus, we consider a 500 m (5 min walk) radius distance for the current study which will cover surrounding areas of both lines, and all indicators will be measured within this area.

### 3.2. Identification of Indicators

Computing TOD-ness involves assessing quantifiable indicators that describe or characterize TOD. These indicators should be suitable for progressive evaluations instead of retrospective assessments. We searched for recent references published between 2017 and 2023 using the Scopus, Google Scholar, and Web of Science referencing databases. These databases were searched for studies that broadly worked on the TOD index or examined the interrelation between multifaceted variables near transit stations or that were transit related in general. Searched results were shortlisted based on the diversity of locations, typology, and citation index. This procedure picked out 20 articles, and a list of common variables is presented in the table below. Variables are broadly classified into different domains like density, land use, and built environment, walkability and cyclability, socioeconomic and demographic, utilization of transit capacity, user friendliness of transport facility, access and accessibility, and parking at the station. Table 2 lists the relevant indicators for TOD planning, extracted from the recent literature (2018–2023) and suggested by various authors.

In the TOD literature, density, diversity, and design are emphasized as the most significant dimensions [43]. Later, two new criteria for measuring the built environment were added: destination accessibility and transit distance [11,44–46]. In destination accessibility, we mean the ease of access to the main attractions of a trip. The distance to the nearest transit stop is determined by calculating the shortest route from a residence or workplace. These 5D criteria are generally interconnected, somehow overlapping and reinforcing each other so they must be included in calculations. In addition to development characteristics, a high-quality transit service is essential for TOD. The success of TOD is greatly influenced by the design and quality of the transit service, and a TOD plan may falter if the transit service lacks attractiveness. A high-quality transit system involves providing frequent service along specific routes and schedules to ensure reliability. Furthermore, a seamless and easily navigable transport network with a joint ticketing system, information screens, and basic amenities is necessary to enhance the attractiveness and user-friendliness of the transit system. Similarly, an increase in ridership can be used as a measure of higher TOD values and should be evaluated. Furthermore, parking facilities for cars and bikes at the stations are important for first and last-mile connectivity.

Considering the above discussion, the indicators for this study have been selected upon careful consideration. The final indicators were selected based on their frequent occurrence in literature and further attention to their quality and availability within the analysis area. The criterion of "user-friendliness of transit service" should be omitted as all stations are equipped with basic amenities and information display systems, regardless of their functionality. Comparing stations on this basis is therefore not feasible. Similarly, the safety of travelers at transit stops also affects the choice of transit use. Most stations have a similar design layout, offering good visibility and lighting, making this indicator inapplicable in our case. Additionally, it is widely recognized that a variety of land uses enhance the vibrancy and security of a location by increasing the presence of people in that area [47].

The frequency of the transit service affects access to transportation and can be computed depending on the number of fleets/trains working per hour, at each station. In our case, the Green line is BRT while the Orange line is LRT, each with different capacities and frequencies. Subsequently, we used this information to calculate Passenger load for every transit node but different stations are not compared for this in TOD calculations.

Although parking for bikes and bicycles is available near transit stations, it is not in the scope of PMA so we could not obtain data regarding demand and supply for that. Some stations also have parking for cars but this indicator had to be removed because there were no authentic data available for it. One important indicator is "Interchange to different routes of same transit" which measures the accessibility of bus/train multiple routes at each station. More bus/train routes and connecting destinations can encourage people to choose transit over driving. This indicator is not relevant in our case, as we are specifically considering two operational lines within the system: the Green line, which is a Bus Rapid Transit (BRT) line, and the Orange line, which is a metro train. Therefore, this aspect may be assessed in the future when the proposed two lines are also constructed.

Table 3 below displays the final selected variables for this research and their brief description and data source. As stated in the below Table, 10 indicators were computed in ArcGIS (10.8) to assess the TOD index within a 500 m buffer area around the 26 train and 27 BRT stations along two corridors. Section 5 provides comprehensive calculations for each indicator, detailing their respective methods and formulas for computation.

**Table 2.** List of indicators from systematic literature review.

| Criteria | TOD Indicators | [48] | [13] | [37] | [49] | [50] | [51] | [52] | [15] | [53] | [11] | [33] | [54] | [35] | [55] | [56] | [44] | [27] | [57] | [58] | [59] |
|---|---|---|---|---|---|---|---|---|---|---|---|---|---|---|---|---|---|---|---|---|---|
| | Case study area | Tehran, Iran | Shanghai, China | Dehli, India | Dehli, Bengaluru, Bhopal, Mumbai, India | China City-level examination | Kufa City, Iraq | Thessaloniki, Greece, | Addis Ababa, Ethiopia | Depok City, Indonesia | Arnhem-Nijmegen Netherlands | Ahmedabad, India | New York, USA | Wuhan, China | Shanghai, China | Beijing, Shanghai, Shenzhen, Wuhan, Hangzhou | Delhi, India | Ningbo, China | Singapore | Dhaka, Bangladesh | Kuala Lumpur, Malaysia |
| | No. of transit stations(cases) | 22 | 333 | 47 | | 37 Chinese cities | 3 | 32 | 22 | 3 | 21 | 7 | 472 | 21 | 347 | | 47 | 63 | 23 | 17 | 69 |
| | Buffer Distance | 500 | 800 | | | 800 | 800 | 400–800 | 800 | 800 | 800 | 200 | 800 | 400 | 500 | | 800 | | 500 | 800 | 800 |
| **Density** | Population density | * | | * | * | * | * | * | * | * | * | * | * | | * | * | * | * | | * | * |
| | Commercial density | * | | * | * | * | * | * | * | * | | | | | | | | | | * | * |
| **Land use and built environment** | Land use diversity | * | * | * | * | * | * | * | * | * | * | * | * | * | * | * | * | | * | * | * |
| | Block Size | | | | | * | | | | | | * | | * | | | | | * | | |
| **Walkability and Cyclability** | Land use Mixedness | * | * | | * | * | | * | * | | * | | * | * | * | * | | * | * | * | * |
| | The total length of walkable/cyclable paths | | * | * | * | | | * | * | * | * | * | | | * | * | | | * | * | * |
| | Intersection density | * | | * | | * | * | * | * | * | * | * | | | * | * | * | * | * | * | * |
| | Impedance pedestrian catchment area (IPCA) | | | | | | | | | * | * | | | | | | | | | | |
| **Economic Development** | The density of business establishments | * | | | * | | | * | | * | * | | * | | * | | | * | * | | |
| **Utilization of Transit Capacity** | No. of entry and exits | | * | | | | | | | | | | * | | * | * | | * | * | | |
| | Passenger load in peak hours | | | | | | | * | | * | * | | | | | | | | | | |

**Table 2.** *Cont.*

| | | [48] | [13] | [37] | [49] | [50] | [51] | [52] | [15] | [53] | [11] | [33] | [54] | [35] | [55] | [56] | [44] | [27] | [57] | [58] | [59] |
|---|---|---|---|---|---|---|---|---|---|---|---|---|---|---|---|---|---|---|---|---|---|
| **Utilization of Transit Capacity** | Passenger load in off-peak hours | | | | | | | * | | * | * | | | | * | | | | | | |
| **User-friendliness of transit facility** | Safety of commuters at the transit stop | | | | * | | | * | | * | * | | | | | | | | | | |
| | Basic Amenities at the station | | | | | | | * | | | | | | | | | | | | | |
| | Presence of information display systems (Yes/no) | | | | | | | * | | * | * | | | | | | | | | | |
| **Access and Accessibility** | Frequency of transit service | | | | | | | | | * | * | | | | * | * | | | | | |
| | Interchange to different routes of the same transit | | * | | | | | * | | * | * | | | | | | | | | | |
| | Interchange to other transit modes | | * | | * | | | * | | * | * | | | | | | | | | | |
| | Access to opportunities within a walkable distance from the train station | * | * | * | | | | * | | * | * | | | * | | * | * | | | | |
| **Station Parking** | Parking utilization for cars/4-wheelers | | | | * | * | | * | * | * | * | * | * | | * | | | | | * | * |
| | Parking utilization for cycles | | | | * | * | * | * | * | * | * | | * | | | | | | * | * | * |
| | Open areas/green spaces | | | | * | | | * | | | | | * | | | | | | * | * | * |

* Mean Yes, while the blank mean No.

**Table 3.** Criteria and indicators for measuring TOD index.

| Sr. No. | Criteria | Icon | Indicators | Description | Data Source |
|---|---|---|---|---|---|
| 1 | Density | | Population density PD | Number of people/sqkm | WorldPop |
| 2 | Land use diversity | | Land use diversity LDI | Entropy index | Calculated using GIS |
| 3 | Walkability and cyclability | | Land use mixedness (MI) | Mixedness of residential land use with other land uses | Calculated using GIS |
| | | | Intersection density ID | Number of intersections/sqkm | Calculated using GIS |
| | | | Walkable/cyclable paths (WCP) | Total length of walkable/cyclable paths (meters) | Calculated using GIS |
| 4 | Economic development | | Business density BD | Number of business establishments/sqkm | Urban Unit |
| 5 | Capacity utilization of transit | | Passenger load in peak hours (PLPH) Passenger load in off-peak hours (PLoPH) | Ridership data per station Transit frequency and capacity | Punjab Masstransit Authority (PMA) |
| 6 | Access and accessibility | | Interchange to other transit modes (IoTM) Job Density (JD) | Interchange to other transit modes; access to opportunities within walkable distance from the transit station (No. of jobs/sqkm) | Survey (2023) Urban Unit |

### 3.3. Calculation of TOD Index

A TOD index will calculate various spatial indicators and sum them under the SMCA framework to obtain a comprehensive value showing the existing levels of TOD around every transit node. High levels of TOD indicate that the urban development characteristics of that node are suitable for the use of transit, which can further help to create justified planning decisions to support location decision making.

3.3.1. Standardization of Indicators

The first stage in the spatial multi-criteria Analysis model is to standardize indicator values for comparison and combination. Standardization of indicators is essential since they are computed using various units of measurement. There are several methods in the literature, including U-shape, maximum, interval, goal, concave, convex, and curve, that can be used to convert the original indicator values into standardized values between 0 and 1. In this study, the researcher employed the "maximum" method, which involves assigning a value of "1" to the highest attained indicator value, and proportionally assigning values between 0 and 1 to all other values based on their ratio to the maximum value. The "benefit" and "cost" representations were combined to assess the "land use mixedness". This indicator functions as a "benefit" to the index until it reaches a value of 0.5, signifying an equilibrium between residential and other types of land uses. Beyond this point, it works as a "cost" and negatively contributes to the TOD index [7].

### 3.3.2. Weighting the Indicators

The process of weighing criteria in decision making is subjective and may vary based on the stakeholder, setting, or time. Researchers faced difficulties in creating a user-friendly decision support system that could incorporate input from various stakeholders with different disciplinary backgrounds [60]. Biased decision making is a significant challenge associated with stakeholder participation. In Sara et al.'s study [40] on Egypt's potential TOD index, two scenarios were tested to assess the impact of changes in criteria weight on the index value. The first scenario utilized indicators and weights from a stakeholders' workshop conducted by Singh et al. [11], though this reference study context is entirely irrelevant. Then, the second scenario assigned equal weight to all indicators to reduce bias. Results showed minimal variation in index results for top-scoring stations but significant improvement for low-scoring stations. In this context, it is pertinent to mention a study conducted by Teklemariam et al. [15] focusing on the potential enhancement of TOD in Addis Ababa, Ethiopia but criteria weighting is missing. In the Lahore case, political, economic, and regional cultural disparities are pivotal considerations. The mass transit lines are politically motivated, initiated by the ruling party PMLN and later halted by PTI in subsequent terms. These projects encountered massive criticism for their substantial budget allocations and perceived limited benefits. So, we have proposed an "Equal weight Scenario" (Table 4) in which all indicators were assigned equal rank and weight, to mitigate the potential influence of bias on the results. After conducting SMCA, a sensitivity analysis was carried out to evaluate the results' strength and to determine the impact of slight weight adjustments on the final TOD index.

**Table 4.** Criteria, indicators, and their weights.

| Sr. No. | Criteria | Equal Weight Scenario | Indicators | Contribution to Criteria | Standardization Method |
|---|---|---|---|---|---|
| 1 | Density | 0.166 | Population density | 100% | Maximum |
| 2 | Land use diversity | 0.166 | Land use diversity | 100% | - |
| 3 | Walkability and cyclability | 0.166 | Land use mixedness | 33.3% | Goal |
| | | | Intersection density | 33.3% | Maximum |
| | | | walkable/cyclable paths | 33.3% | Maximum |
| 4 | Economic development | 0.166 | Business density | 100% | Maximum |
| 5 | Capacity utilization of transit | 0.166 | Passenger load in peak hours | 50% | - |
| | | | Passenger load in off-peak hours | 50% | - |
| 6 | Access and accessibility | 0.166 | Interchange to other transit modes | 50% | - |
| | | | Job density | 50% | Maximum |

## 4. The Study Settings

### 4.1. Background Study of Lahore

The case study area selected for this research is Lahore city, the cultural capital of Pakistan, which is one of the world's most populous cities and serves as a major hub for economics, politics, transportation, entertainment, and education. As of the 2017 census [61], Lahore's population was recorded at 11,126,285, with an average annual growth rate of 4.07% since 1998, and, as an economic hub, it also assists 13.2% of the national GDP and is rated 122nd biggest in the world for GDP [62]. The historical city of Lahore, with an extensive history of over 2000 years, is enhanced with splendid architectural and urban heritage combined with a thriving and booming cultural life [63].

Previously, Lahore was under the rule of the Mughal Empire and later the British in the Indian Subcontinent. Following Pakistan's independence in 1947, it turned into a part of the Islamic Republic. Its historic structures, mosques, temples, churches, tombs, and public parks make it an amazing tourist destination. Lahore has evolved to include the densely populated walled city, as well as the less dense adjacent urban and suburban areas to the south and southeast [64]. It is a bustling urban area with diverse business opportunities and a growing technology industry.

Lahore was once called the "Mughal city of gardens" [65] and "Paris of the East". However, the city has changed in recent years due to urban renewal, restoration works, and adamant infrastructural expansion. It has transformed from the city of gardens to the "city of concrete", facing several issues like urban decay and land price escalation [66].

Many green spaces in the city are transformed into paving during flyovers, underpasses, and road widening with little respect for the urban green belts. The construction of the Green line and Orange line also damages the environment and cultural image of the city. Lahore is much developed regarding infrastructure [67]; still, it fails to address many issues as these large-scale infrastructure projects are developed without a comprehensive plan or strategy in place [68]. Lahore is experiencing major urban transformations under the newly built rapid mass transit system. which forms the basis of this research to develop a joint development policy to retain its cultural identity while moving towards transit-oriented urban expansion.

There is some amount of TOD in every neighborhood in a city [69]. Where there is sufficient demand for transport, transit facilities follow the urban layout, which generates transit-adjacent development (TAD) that does not interact with it. An ineffective (re)orientation persists in the surroundings as TADs [11]. So, TOD is not just an urban structure close to transit service, the link between them is more important. A TOD is created when the urban structure is furthermore (re)oriented toward transit amenities.

Recognizing this issue, TOD policies serve as an addendum to the planning process, designating specific catchment zones around transit stations/corridors as areas for special development with distinct regulations from the rest of the city.

### 4.2. The Lahore Rapid Mass Transit Rail Project (LRMTRP)

Public transportation in Lahore can be broadly classified into two categories:

- Mass transit system (Green and Orange line) and feeder bus service, run by Punjab Masstransit Authority (PMA);
- Paratransit service (auto and Qingqi rickshaws) privately owned.

The Lahore Rapid Mass Transit Rail Project (*LRMTRP*) [70] (See Figure 4) includes establishing 97 km rail lines to provide sustainable transportation in Lahore to reduce traffic congestion and increase auto dependence. The project was divided into two stages: Phase 1 includes the Green line and Orange line, while Phase 2 includes the Blue and Purple lines as follows:

- Green line (BRT)—Ferozepur Road/Mall Road/Ravi Road/Shahdara (Launched as BRT, operational since 10 February 2013)
- Orange line (MRT)—Raiwind Road/Multan Road/Macleod Road/Railway Station/GT Road (operational since 25 October 2020)
- Blue line (MRT)—Township/Gulberg Boulevard/Jail Road
- Purple line (MRT)—Bhatti Gate/Allama Iqbal Road/Airport

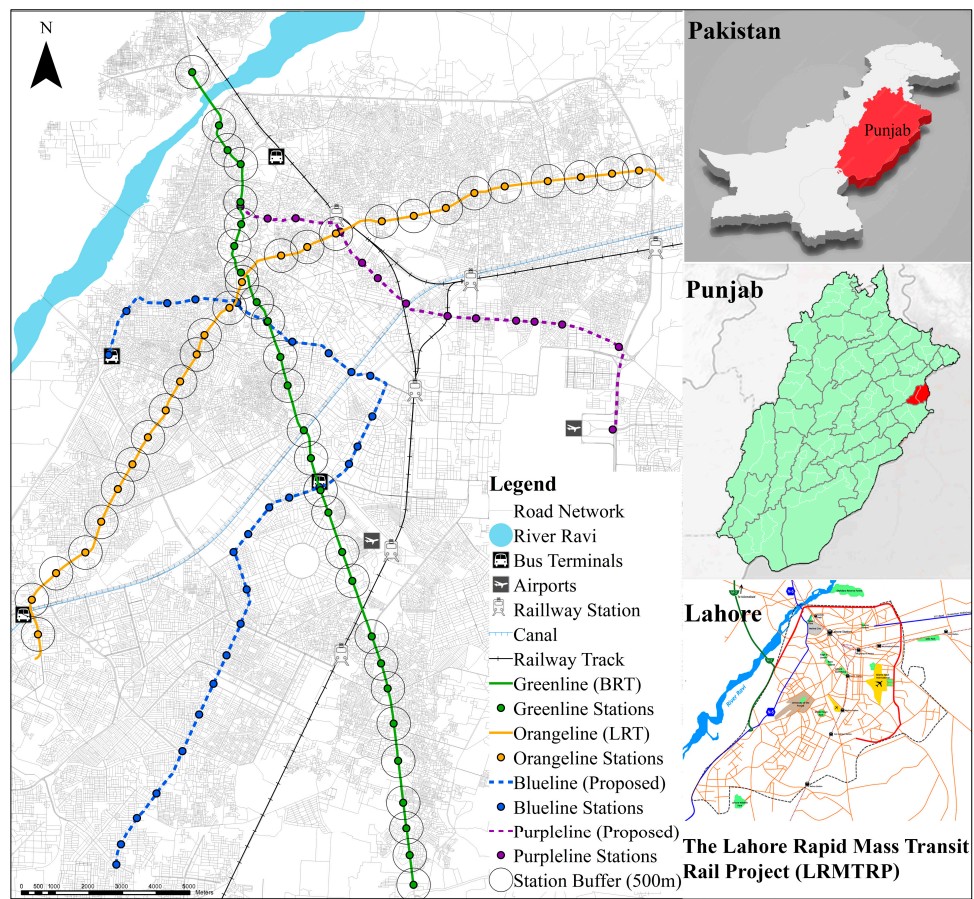

**Figure 4.** The Lahore Rapid Mass Transit Rail Project (LRMTRP). Source: Author.

### 4.2.1. The Metro Bus System (MBS) Lahore (Green Line)

This is Pakistan's initial rapid mass transit project, covering a 27 Km route from north to south through the city center. Within this corridor, 19 km is at grade, and an 8 Km line segment is elevated. The route covers several residential and commercial areas through Ferozepur Road, Lytton Road, Jain Mandar, MAO College, Lower Mall, Civil Secretariat, Aiwan-i-Adal, Katchehry Chowk, Data Darbar, Niazi Chowk, Ravi Road, and Shahdara Town, as shown in Figure 5.

MBS Lahore's construction began in March 2012, and the buses started operating in February 2013. Native and Turkish experts were involved in designing the project. The Traffic Engineering and Planning Agency (TEPA) was responsible for its construction along with the Lahore Development Authority (LDA) for 11 million USD per kilometer [71]. The system was constructed through a partnership between the Punjab Government and the Turkish Government under the build-operate-transfer (BOT) policy [72].

The MBS corridor is a 10 m wide barrier-controlled dedicated corridor designed as a two-lane undivided busway. It has 27 bus stations designed with two curb-side platforms and nine pedestrian bridges along the route. Most stations provide a parking space for bicycles and motorbikes, while some have car parking facilities. Each platform is 81 m long and 3.5 m wide, with three bus bays to stop three articulated buses simultaneously. The platforms can be accessed via escalators, stairs, pedestrian bridges, or an underground passageway through turnstiles as a gated entry and exit. The system operates 64 articulated buses with a top speed of 50 km/h, achieving a commercial speed of 26 km/h with approximately 2 min headways. Commuters are facilitated with level boarding and can enter the bus through four platform sliding doors (PSD) at each bus bay. The system is designed to be controlled and checked through the control center established at the Arfa Software Technology Park, Lahore. To improve the efficiency and reliability of the system,

the Intelligent Transportation System aids the Metro buses to receive signal priority at eight signalized intersections along the bus corridors. The Passenger Information System (PIS) provides real-time bus information at stations and on buses [73].

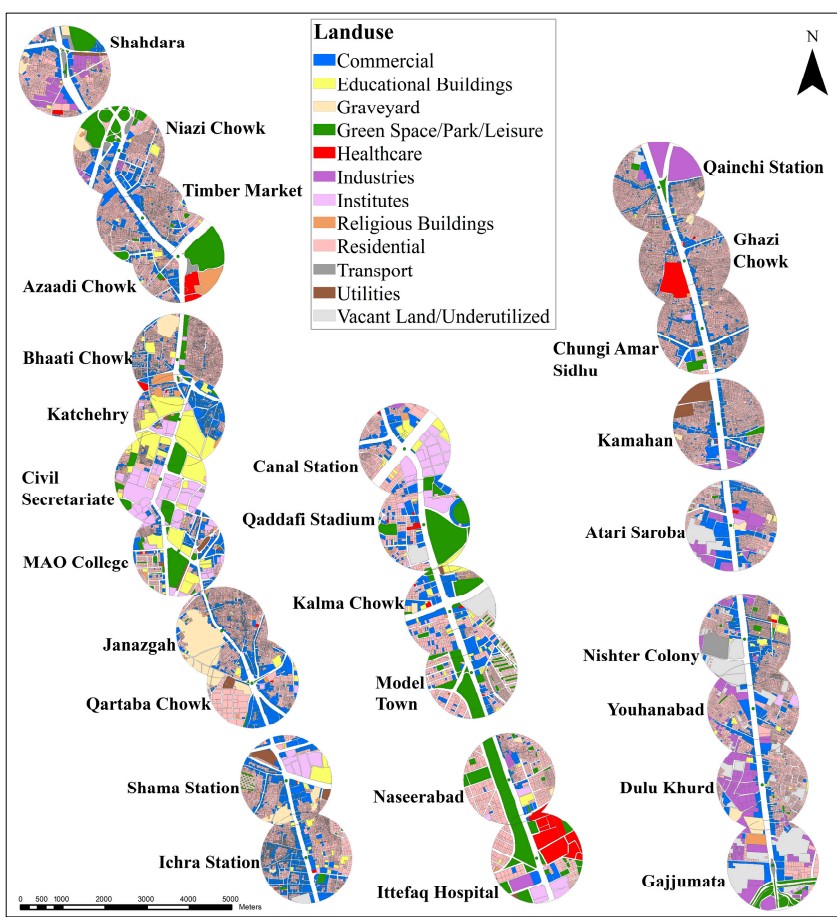

**Figure 5.** Land use distribution around the Green line. Source: Author.

4.2.2. Orange Line

The second line of Phase 1 is the orange line, a Metro train with a 27.1 km long route in the east–west direction; among that, 25.4 km is elevated, and 1.72 km is underground. Its daily ridership is 245,000 passengers [73] working with 27 train sets (5 cars/train set). The OLMTS route passes through the area where 26 cultural and archaeological buildings and sites are located. The route is also significant as it connects various important nodes in Lahore like Shalimar Gardens, the University of Engineering and Technology (UET), Railway Station, Chauburjii Square, and Ali Town (See Figure 6). Moreover, its counterparts' various famous roads in Lahore comprise GT Road, MacLeod Road, Multan Road, and Raiwind Road. The Orange and Green lines cross each other at Anarkali, so the Orange line passengers will utilize the Anarkali Transfer Station to shift to the Green line (MBS) at the MAO college station.

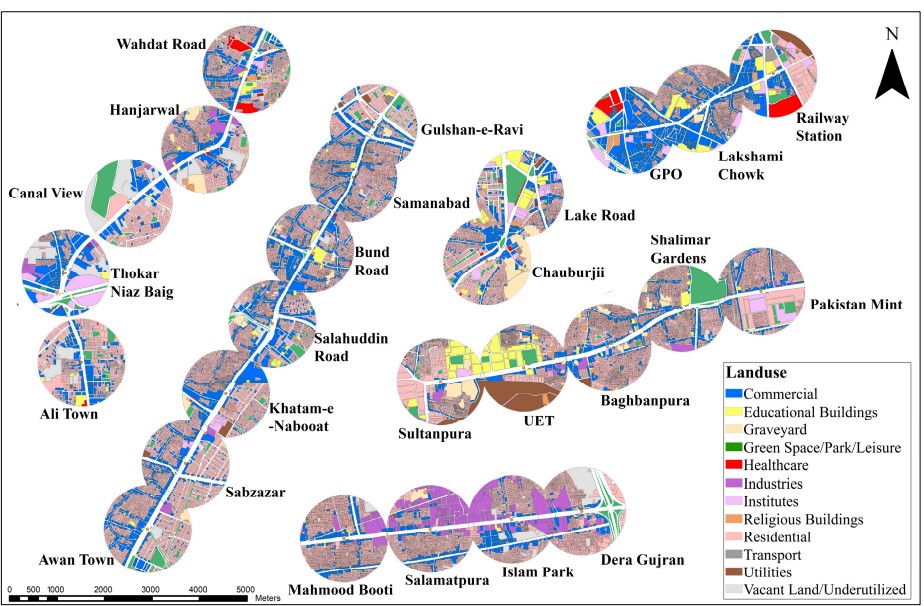

**Figure 6.** Land use distribution around Orange line. Source: Author.

### 4.2.3. Blue Line

The Blue line of Phase 2 will have a 27.6 km track (20.95 km elevated and 6.65 km underground), starting from Valencia Town and terminating at Babu Sabu. It will have 28 stations (22 elevated and 6 underground), and the portion from Lahore College for the Women University to Babu Sabu will be underground (6.65 km) [73].

### 4.2.4. The Purple Line

The Purple line will have a 16.07 km route (11.46 km elevated and 4.61 km underground) and 14 stations (9 elevated and 5 underground), originating from Allama Iqbal International Airport in the south and terminating at Bhatti Chowk traversing through the major intersections of Guldasht Town, Jora Pull, Askari IX, Saddar Chowk, Daharmpura, Garhi Shahu, Railway Colony, Railway Station, Brandreth Road, and Shah Alam Market. The portion from Bhatti Chowk to Garhi Shahu is underground (4.61 km), and this line is also known as the Airport line [73].

## 5. Results Derived from Indicator Calculations

### 5.1. Population Density

The literature often emphasizes the importance of high population densities in supporting more frequent transit services, increasing transit ridership, and creating vibrant, pedestrian-friendly communities. Therefore, high-density development indicates a high demand for travel, and consequently, high levels of transit-oriented development (TOD) in an area.

It is a measurement of population per unit of land area (Equation (1)). The WorldPop open population repository (WOPR) is used to obtain a gridded population count dataset for Pakistan for the year 2020 (100 m resolution). The dataset is available in Geotiff format, which was further processed in ArcGIS to calculate population density in every buffer area (Figure 7a).

$$Population\ Density = \frac{\text{No.of People}}{\text{Land Area}} \tag{1}$$

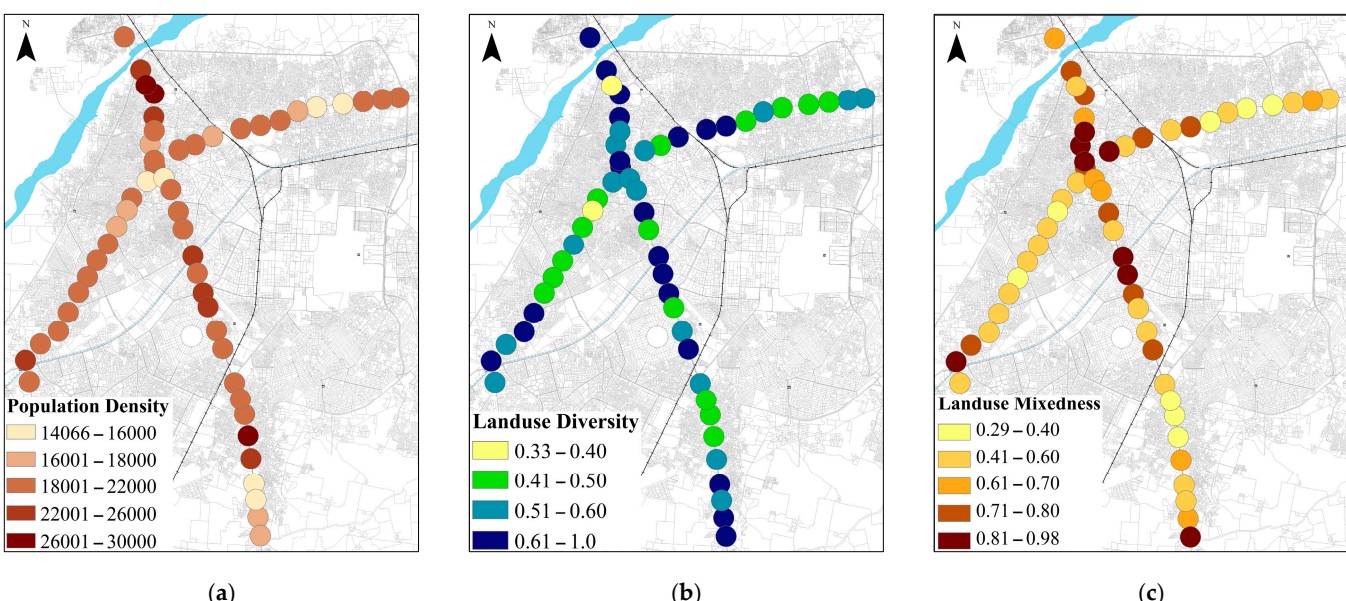

**Figure 7.** (**a**) Population density, (**b**) land use diversity, (**c**) land use mixedness around the Green line and Orange line (500 m buffer).

*5.2. Land Use Diversity*

Land use diversity evaluates the degree of spatial diversity by examining the distribution of different land use types within a walkable area. These diverse land use types of an area fulfill the several needs of its residents, which ultimately enhance transport utilization in off-peak hours and on holidays. This study employed the "entropy method" to compute the diversity of land use among various approaches available, as follows in the formula (Equation (2)):

$$LU_d(i) = \frac{-\sum_i Q_{lu_i} \times \ln(Q_{lu_i})}{\ln(n)}$$ (2)

where

$Q_{lu_i} = \frac{S_{lu_i}}{S_i}$

$LU_d(i)$ land use diversity in the area of analysis = (*i*)

$lu_i$ = land use class (1,2,...,*n*) within analysis area *i*

$Q_{lu_i}$ = Share of specific land use with in analysis area *i*

$S_{lu_i}$ = Total area of the specific land use with in the analysis area *i*

$S_i$ = Total area of analysis *i*

Parcel level land use data (vector data) were formulated in ArcGIS 10.8 for a 500 m buffer area around every station with the help of surveys, Google Earth, an open street map, and multiple site visits. The land uses included for this study were commercial, educational, health, institutes, industries, religious buildings, green/parks/leisure spaces, graveyards, underutilized/vacant land, residential, utilities, and transport. Figure 8 illustrates the current land use surrounding a single station. The average distance between stations is around 1 km so both corridors are covered with a 500 m buffer area. The resulting index is between 0 and 1, where higher values indicate greater levels of land use diversity and therefore better levels of TOD-ness. Figure 7b demonstrates the variation in land use diversity along the Green line and Orange line.

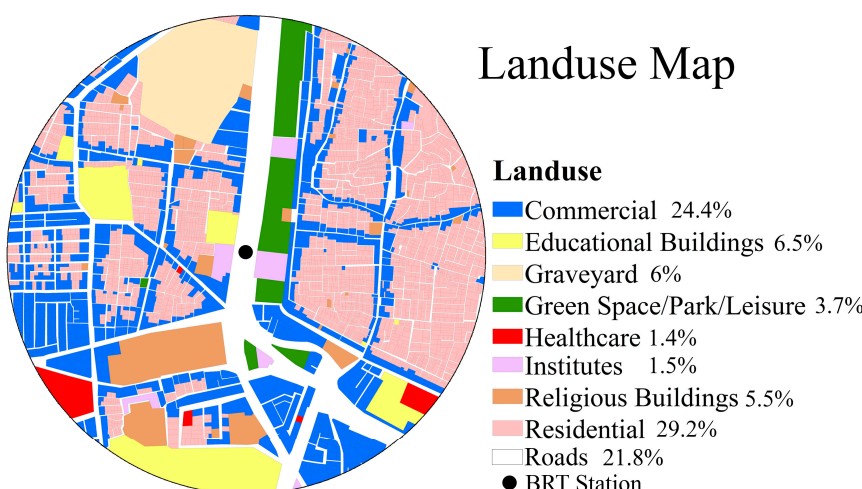

**Figure 8.** Existing land use around Bhaati Station.

### 5.3. Land Use Mixedness

The purpose of this indicator is to evaluate the area's suitability for walking and cycling. It differs from the land use diversity as it specifically assesses the mixing of residential land use with other types of land use. A sufficient mix of residential land use with other land uses enables many non-work trips which can be possible on foot or bicycle, as suggested by many authors [8,48]. So, we measured mixedness using the following formula as developed by Zhang and Guindon [74] (Equation (3)):

$$MI_{(i)} = \frac{\sum_{ni} S_c}{\sum_{ni}(S_c + S_r)} \forall i \text{ a}$$

(3)

where

$MI(i)$ = "mixedness index" for analysis area $(i)$
$S_c$ = sum of the total area under non − residential urban land uses
$S_r$ = sum of the total area under residential land use within $(i)$

The value of the mixedness index (MI) can vary from 0 to 1, where 0 means the absence of residential land use and 1 means the area is exclusively residential. A mixedness index of 0.5 indicates an equal distribution of residential and other land uses, reflecting a balanced mix. Figure 7c demonstrates the variation in land use mixedness along the Green line and Orange line.

### 5.4. Intersection Density

Intersections can facilitate walking and cycling by reducing the distance of travel. Higher intersection density means higher walkability or cyclability because of choices available to shorten their routes [8,43]. The road network datasets were used in ArcGIS (10.8) to calculate the intersection density which is represented by the number of intersections per km$^2$ in every station buffer area (Figure 9).

### 5.5. Total Length of Walkable/Cyclable Paths

This indicator is calculated using the length of accessible roads for pedestrians and cyclists within the buffer area around each transit node, measured in meters (Figure 10).

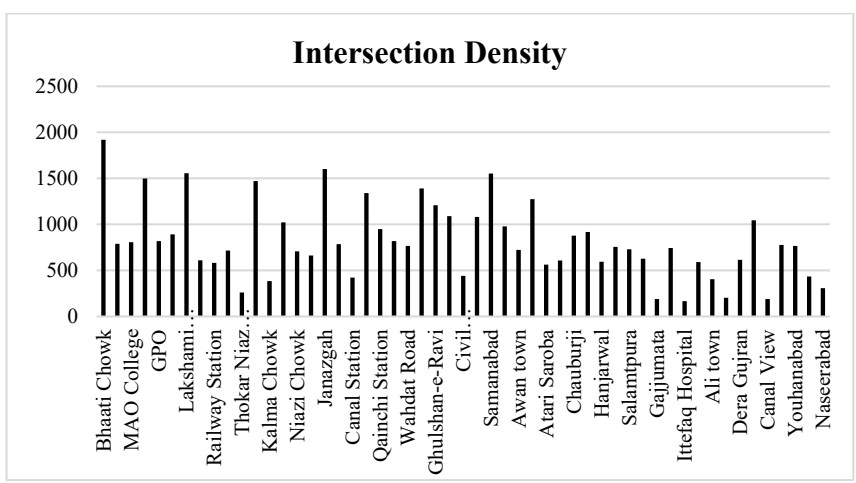

**Figure 9.** Intersection density around Green line and Orange line (500 m buffer).

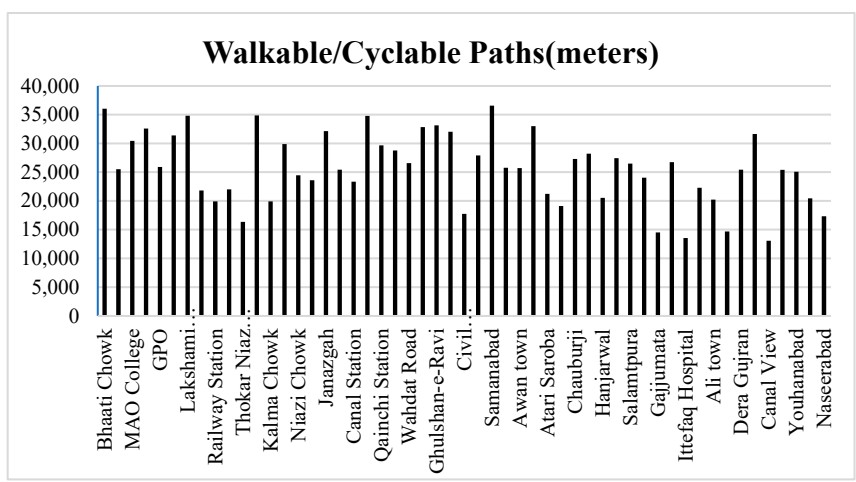

**Figure 10.** The total length of walkable and cyclable paths around Green line and Orange line (500 m buffer).

*5.6. Business Density*

In the land use classification, commercial establishments encompass both retail services and non-retail enterprises, including consultancy businesses. So, business density is calculated as the number of business establishments/sqkm within the designated area for each station's analysis.

*5.7. Passenger Load*

Transit ridership serves as a measure of TOD-ness, with increased ridership correlating to higher levels of TODs. The utilization of the transit's capacity should be measured by evaluating ridership at each node. It can be calculated through Passenger load which is the total number of passengers divided by the total capacity of the transit system at that station. A greater Passenger load indicates improved utilization of transport capacity and a better level of TOD.

For this indicator, relevant data were acquired from the Punjab Masstransit Authority (PMA) as both lines are working under it. Basic information regarding no. of fleet/train operating, fleet/train capacity, and operational hours was available on their website. Other information including the total number of passengers per station per day and total trips per day was obtained from PMA specifically. We need to calculate the Passenger load for peak and off-peak hours, so we have to properly divide the daily total. But, in the absence

of specific split data, it was assumed that 25% of the daily traffic occurs during peak hours and 75% occurs during off-peak hours at every node.

Capacity calculations took into account the frequency of service during peak and off-peak hours. The Green line is a bus-based system while the Orange line is a rail-based system that serves Lahore city and the capacity of each is different. The total capacity of the transit system for every station was computed through their fleet/train capacity and their operational hours. We also took into account that buses/trains are not vacant upon reaching a station, except if they originate from a terminus station. Therefore, we assumed a 40% occupancy for non-terminus stations. Shahdara and Gajumata are terminal stations of the Green line, while Ali town and Dera Gujran are terminus stations of the Orange line. Considering all relevant aspects, we calculated the total capacity of the transit system during peak and off-peak hours. Ultimately, the Passenger load for each station was calculated (Figure 11).

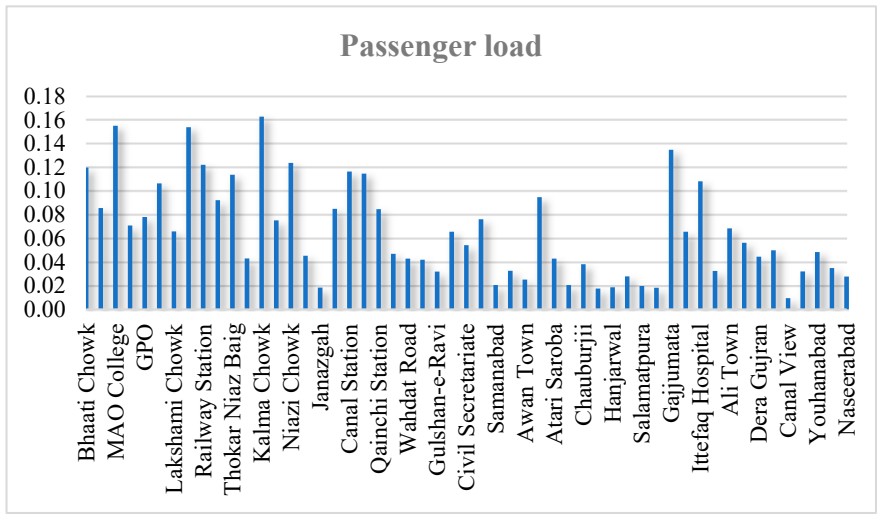

**Figure 11.** Passenger load for Green line and Orange line.

*5.8. Interchange to Other Transit Modes*

A good quality transport system provides access to numerous destinations and allows for the interchange between different modes of transportation. In Lahore, with its main railway station and three intra-city bus terminals, we assessed the interchange options at each station, using a value of 0 to indicate no interchange and a value of 1 to signify interchange availability at an off-street bus stop (Table 5).

**Table 5.** Interchange stations.

| Station Name | Line | Interchange Options |
|---|---|---|
| MAO college | Green line | Orange line (LRT) |
| Lake road | Orange line | Green line (BRT) |
| Railway station | Orange line | Lahore Junction railway station |
| Kalma chowk | Green line | Daewoo bus service |
| Thokar Niaz Beg | Orange line | Jinnah bus terminal, Daewoo bus service |
| Azadi Chowk | Green line | Lahore Badami Bagh bus terminal |
| Bund Road | Orange line | Lahore city bus terminal |
| Shahdara | Green line | Shahdara bus terminal |

Thokar Niaz Beg is a major entry and exit point from south Lahore. The Lahore Jinnah bus terminal, also known as the Thokar Niaz Beg bus terminal, is located at the interchange between the M2 Motorway and Multan Road. Transit riders can use Thokar Niaz Beg station of the Orange line as an interchange to reach this bus terminal. The Daewoo bus

service is also situated at Thokar Niaz Beg; their one terminal is located near Kalma Chowk. So, transit riders can use the Kalma Chowk station of the Green line to reach this terminal. The Lahore Badami Bagh bus terminal, also known as Lari Adda, is a major entry or exit point for all bus travelers. It is among the three major bus terminals of Lahore city and is situated near Badami Bagh railway station on a circular road. Transit riders can use the Azadi Chowk station of the Green line as an interchange to it. The Lahore City bus terminal, also known as Niazi Adda, is located on Bund Road. Many famous bus services are available at this terminal. Transit riders can use the Bund Road station of the Orange line as an interchange to this terminal. The "Railway station" of the Orange line is used as an interchange station to the Lahore Junction railway station which is a major transport hub for local and intercity trains. The Shahdara station of the Green line can be used as an interchange with the Shahdara bus terminal.

### 5.9. Access to Opportunities

This indicator evaluates the number of job opportunities available within the area of analysis for each station.

$$\text{Job density} = \text{No. of jobs/sqkm} \tag{4}$$

## 6. Results and Sensitivity Analysis

The TOD index result is crucial for understanding the characteristics of an area as "TOD". The TOD index results for all 53 stations in Lahore city are tabulated in Table 6 and shown in Figure 12. Based on these results, we can compare station areas and identify the ones with the highest and lowest levels of TOD in the city. In the absence of a benchmark, when using the same inputs within a specific time frame, for a similar objective and evaluating a similar group of objects, the results are comparable within the group itself. As this is the initial attempt to evaluate the level of transit-oriented development (TOD) for transit lines in a Pakistani city, there is no previous study benchmark available. Therefore, for better understanding, we can compare all the stations with each other.

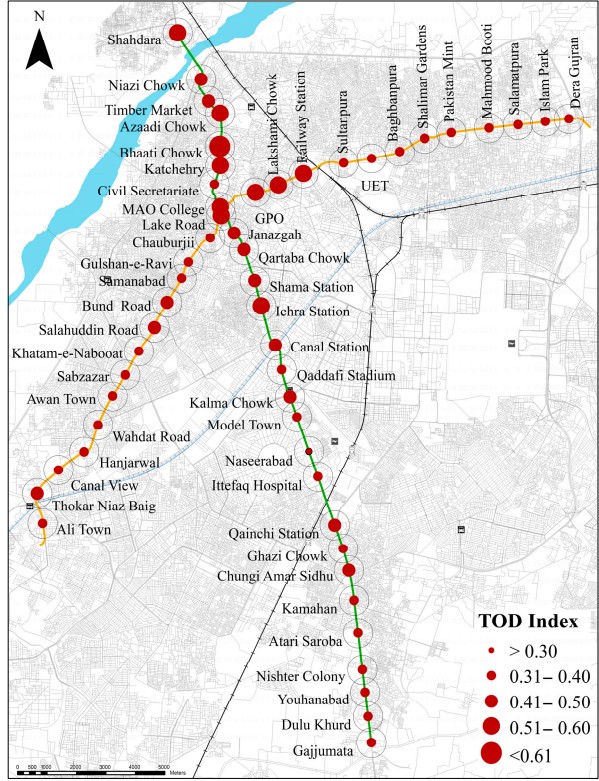

**Figure 12.** TOD index for all 53 stations around the Green line and Orange line in Lahore.

**Table 6.** Criteria and TOD index values for all 53 stations around the Green line and Orange line in Lahore.

| S. No. | Station Name (Ranked by Their TOD Score) | TOD Index Score | Criteria | | | | | |
|---|---|---|---|---|---|---|---|---|
| | | | Density | Land Use Diversity | Walkability and Cyclability | Capacity Utilization of Transit | Economic Development | Accessibility |
| 1 | Bhaati Chowk | 0.61 | 0.82 | 0.65 | 0.87 | 0.12 | 1.00 | 0.27 |
| 2 | Azaadi Chowk | 0.60 | 1.00 | 0.67 | 0.62 | 0.09 | 0.53 | 0.76 |
| 3 | MAO College | 0.54 | 0.66 | 0.72 | 0.69 | 0.16 | 0.30 | 0.81 |
| 4 | Ichra Station | 0.54 | 0.72 | 0.56 | 0.74 | 0.07 | 0.93 | 0.32 |
| 5 | GPO | 0.53 | 0.69 | 0.68 | 0.68 | 0.08 | 0.74 | 0.50 |
| 6 | Lake Road | 0.52 | 0.64 | 0.76 | 0.68 | 0.11 | 0.29 | 0.76 |
| 7 | Lakshmi Chowk | 0.52 | 0.79 | 0.67 | 0.77 | 0.07 | 0.75 | 0.33 |
| 8 | Shahdara | 0.51 | 0.76 | 0.67 | 0.52 | 0.15 | 0.25 | 0.82 |
| 9 | Railway Station | 0.51 | 0.58 | 0.73 | 0.52 | 0.12 | 0.35 | 0.81 |
| 10 | Katchehry | 0.51 | 0.68 | 0.59 | 0.63 | 0.09 | 0.64 | 0.46 |
| 11 | Thokar Niaz Baig | 0.50 | 0.81 | 0.49 | 0.49 | 0.11 | 0.28 | 0.70 |
| 12 | Timber Market | 0.50 | 0.96 | 0.47 | 0.73 | 0.04 | 0.68 | 0.26 |
| 13 | Kalma Chowk | 0.50 | 0.87 | 0.49 | 0.49 | 0.16 | 0.13 | 0.75 |
| 14 | Bund Road | 0.49 | 0.62 | 0.65 | 0.61 | 0.08 | 0.56 | 0.64 |
| 15 | Niazi Chowk | 0.48 | 0.83 | 0.67 | 0.60 | 0.12 | 0.47 | 0.26 |
| 16 | Shama Station | 0.46 | 0.74 | 0.72 | 0.58 | 0.05 | 0.35 | 0.35 |
| 17 | Janazgah | 0.45 | 0.55 | 0.49 | 0.78 | 0.02 | 0.68 | 0.19 |
| 18 | Qartaba Chowk | 0.44 | 0.64 | 0.56 | 0.58 | 0.08 | 0.57 | 0.32 |
| 19 | Canal Station | 0.44 | 0.80 | 0.73 | 0.56 | 0.12 | 0.18 | 0.38 |
| 20 | Chungi Amar Sidhu | 0.42 | 0.76 | 0.47 | 0.68 | 0.11 | 0.48 | 0.10 |
| 21 | Qainchi Station | 0.41 | 0.77 | 0.67 | 0.61 | 0.08 | 0.36 | 0.15 |
| 22 | Salahuddin Road | 0.41 | 0.72 | 0.76 | 0.58 | 0.05 | 0.47 | 0.13 |
| 23 | Wahdat Road | 0.40 | 0.73 | 0.59 | 0.55 | 0.04 | 0.38 | 0.14 |
| 24 | Baghbanpura | 0.40 | 0.74 | 0.68 | 0.66 | 0.04 | 0.37 | 0.17 |
| 25 | Gulshan-e-Ravi | 0.39 | 0.68 | 0.38 | 0.65 | 0.03 | 0.48 | 0.08 |
| 26 | Kamahan | 0.39 | 0.95 | 0.67 | 0.59 | 0.07 | 0.36 | 0.02 |
| 27 | Civil Secretariate | 0.38 | 0.63 | 0.49 | 0.55 | 0.05 | 0.11 | 0.43 |
| 28 | Shalimar Gardens | 0.38 | 0.56 | 0.64 | 0.60 | 0.08 | 0.36 | 0.19 |
| 29 | Samanabad | 0.38 | 0.59 | 0.70 | 0.70 | 0.02 | 0.60 | 0.09 |
| 30 | Sultanpura | 0.38 | 0.76 | 0.51 | 0.57 | 0.03 | 0.16 | 0.14 |
| 31 | Awan Town | 0.37 | 0.72 | 0.52 | 0.50 | 0.02 | 0.41 | 0.10 |
| 32 | Ghazi Chowk | 0.37 | 0.67 | 0.65 | 0.65 | 0.09 | 0.33 | 0.14 |
| 33 | Atari Saroba | 0.37 | 0.79 | 0.43 | 0.50 | 0.04 | 0.21 | 0.17 |
| 34 | UET | 0.37 | 0.69 | 0.54 | 0.53 | 0.02 | 0.13 | 0.19 |
| 35 | Chauburjii | 0.37 | 0.55 | 0.52 | 0.58 | 0.04 | 0.29 | 0.18 |
| 36 | Khatam-e-Nabooat | 0.37 | 0.74 | 0.62 | 0.56 | 0.02 | 0.32 | 0.11 |
| 37 | Hanjarwal | 0.36 | 0.68 | 0.46 | 0.48 | 0.02 | 0.24 | 0.11 |
| 38 | Sabzazar | 0.36 | 0.74 | 0.46 | 0.50 | 0.03 | 0.40 | 0.09 |
| 39 | Salamatpura | 0.36 | 0.70 | 0.41 | 0.56 | 0.02 | 0.21 | 0.18 |
| 40 | Islam Park | 0.36 | 0.69 | 0.58 | 0.54 | 0.02 | 0.19 | 0.19 |
| 41 | Gajjumata | 0.35 | 0.62 | 0.57 | 0.49 | 0.13 | 0.07 | 0.20 |
| 42 | Nishter Colony | 0.35 | 0.49 | 0.33 | 0.57 | 0.07 | 0.25 | 0.10 |
| 43 | Ittefaq Hospital | 0.35 | 0.74 | 0.65 | 0.41 | 0.11 | 0.00 | 0.22 |
| 44 | Dulu Khurd | 0.34 | 0.59 | 0.5 | 0.53 | 0.03 | 0.10 | 0.22 |
| 45 | Ali Town | 0.34 | 0.72 | 0.42 | 0.42 | 0.07 | 0.24 | 0.10 |
| 46 | Qaddafi stadium | 0.34 | 0.71 | 0.54 | 0.44 | 0.06 | 0.08 | 0.16 |
| 47 | Dera Gujran | 0.34 | 0.75 | 0.67 | 0.52 | 0.04 | 0.09 | 0.10 |
| 48 | Mahmood Booti | 0.34 | 0.55 | 0.59 | 0.59 | 0.05 | 0.31 | 0.12 |
| 49 | Canal View | 0.33 | 0.74 | 0.48 | 0.39 | 0.01 | 0.08 | 0.15 |
| 50 | Pakistan Mint | 0.32 | 0.54 | 0.65 | 0.49 | 0.03 | 0.29 | 0.14 |
| 51 | Youhanabad | 0.32 | 0.50 | 0.42 | 0.55 | 0.05 | 0.16 | 0.13 |
| 52 | Model town | 0.32 | 0.81 | 0.49 | 0.44 | 0.03 | 0.09 | 0.08 |
| 53 | Naseerabad | 0.29 | 0.66 | 0.53 | 0.38 | 0.03 | 0.08 | 0.06 |

In our study, Bhaati Chowk and Azaadi Chowk obtained the highest TOD scores of 0.61 and 0.60, respectively. The lowermost score of 0.29 was noted for Naseerabad station. It can be noticed that TOD levels are mostly low in the entire region. An overview of the scores for the individual indicators shows "Capacity utilization of transit" is scoring very low around all stations (0.01–0.16). Station-wise ridership data for the last 6 months were obtained from PMA to calculate Passenger load at each station which shows very low utilization of both lines. Even top-scoring stations have values of 0.12 and 0.09 for this criterion, a reason behind their less-favorable score.

Considering the transit statistics, for the Green line, each platform is designed for 3 articulated buses simultaneously and the current system is running with 64 articulated buses with a maximum capacity of 150 passengers (sitting and standing). According to the PMA website, operation hours are 6 am–10 pm which means 16 working hours, and the average number of daily trips is 640, which means its frequency is 40.

Frequency of transit service = no. of buses/trains working/hour.

In this regard, the current system is running with a capacity of 3000 pphd while it is designed for a capacity of 12,000 pphd, which means it is running at 1/4th capacity. Observational surveys have shown that there is only one bus at a time and the other two bays are always vacant. Moreover, buses are always crowded, making it an uncomfortable choice for older people and women with kids. The design and quality of the transit service significantly impact the potential victory of TOD, and TOD planning can falter if the transport system lacks attractiveness.

Based on the Orange line statistics, the system operates with 27 train sets (5 cars per set) with a maximum capacity of 1000 passengers (seated and standing). With an average of 290 daily trips, the system's frequency is 18. Currently, it operates at 60% capacity (9000 pphd) versus its designed capacity of 15,000 pphd. However, the low ridership results in a lower Passenger load, indicating underutilization of transit capacity. Thus, the primary factor contributing to the low TOD index scores for all stations along both lines is the underutilization of transit capacity.

We conducted a sensitivity analysis to assess the reliability of our outcomes and approach in light of potential uncertainties in data and indicator values. We anticipate minimal uncertainty in the data due to the predominant use of primary data and the reliance on well-verified and dependable secondary data sources. Consequently, conducting a sensitivity analysis for the data inputs is deemed unnecessary. However, an assumption was made regarding the peak/off-peak hour split of passenger traffic, which was utilized to calculate the "Passenger load" of transit during peak and off-peak hours. This assumption may introduce potential uncertainty. To assess the sensitivity of the TOD index values to changes in this assumption, the peak/off-peak hour split for Passenger load calculations was adjusted from 25/75 to 20/80. It is noteworthy that the weight assigned to "Passenger load" in the TOD index calculations is 0.166, representing approximately 17%. Upon recalculating the TOD index results for all stations, a slight variation of 0.01 was observed because both transit systems are operating at a low capacity as per the record.

## 7. Inferences and Discussion

This study focused on evaluating the level of TOD to establish a reference for sustainable urban planning and to maximize the utilization of transit services (BRT and LRT) in Lahore, Pakistan. In this segment, we analyze our results, correlate various station areas, and delve into the index outcomes to develop policy recommendations for the entire city and specific stations. The figure shows higher TOD levels at Bhaati Chowk and Azaadi Chowk stations than others, highlighting the concentrated development around these two main centers.

The study area is a multicentric city district with diverse economic, cultural, and administrative centers, leading to the development of location orders in the network. These orders reflect varying development potentials among different stations. So, a station's development level is linked to its position within the overall network. Bertolini (1999)

states that activity concentration naturally happens around areas with high human traffic, such as transit nodes, where diverse individuals with varying travel purposes converge. Such a high and varied volume of public transport stations can result in evenly varied forms of human communications and communal and financial activities. Hence, each station, contingent on its position and transport services, has the potential to drive transit-supportive urban growth or TOD based on its position in the network [75].

The Green line and Orange line in Lahore are priority lines due to route importance, travel demand, and addressing social and environmental challenges. There is a need to work on individual stations to improve TOD scores by improving their low-scoring criteria to achieve sustainability goals from these high-budget projects. Based on the regional score distribution (0.61–0.54), four top-scoring stations (Bhaati Chowk, Azaadi Chowk, MAO College, and Ichra station) have been identified as having a high potential for improvement. Similarly, Pakistan Mint, Youhanabad, Model Town, and Naseerabad are identified as the lowest-scoring stations (0.32–0.29). The current Lahore District Master Plan for 2050, has identified a 500 m buffer area around both lines as a TOD zone for densification and regeneration. In the TOD policy for the Lahore district, the top-scoring transit nodes can be given priority for enhancing the TOD levels in their vicinity. Other transit stations could also be identified in the same way.

It is widely recognized that investment policies, maps, and descriptive plans alone are insufficient to stimulate TOD and must be supplemented with specific actions or tools. Therefore, the web diagrams (see Appendix A) can assist in pinpointing areas for improvement based on low-scoring criteria for each station. The TOD index results enable the identification of distinct issues specific to each station area, which cannot be achieved through clustering techniques. The web diagrams are created using standardized values. For example, a standardized score of 1 for population density at Azaadi Chowk station indicates that its population density is the highest among all stations. Similarly, a maximum score of 0.76 for land use diversity at the railway station means that its diversity is the highest among all stations, but there is still room for improvement. Nonetheless, prioritizing low-scoring criteria for improvement is crucial for enhancing overall TOD. Therefore, web diagrams for the four top-scoring stations and four lowest-scoring stations can elucidate the trend in variation across all criteria, along with suggested improvements (Figure 13).

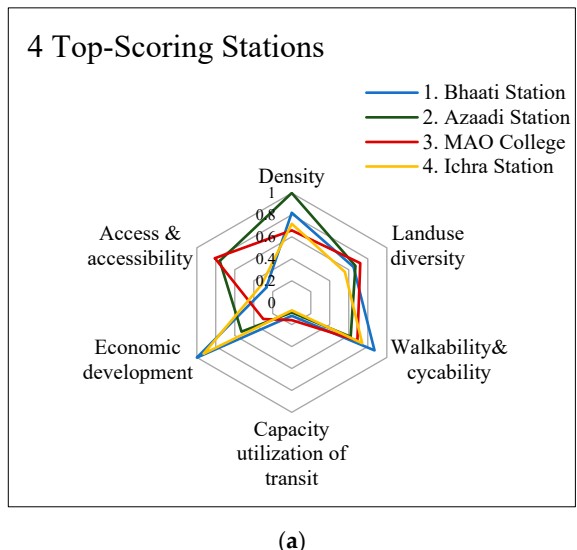
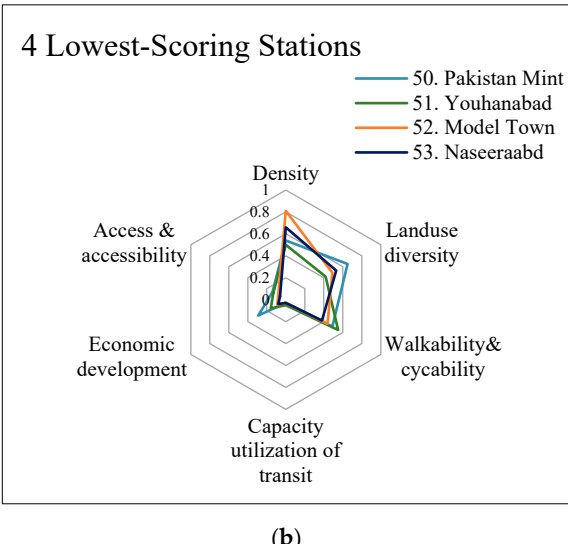

(**a**)            (**b**)

**Figure 13.** (**a**) Web diagram for the four top-scoring stations, (**b**) web diagram for the four lowest-scoring stations.

Web diagrams were employed to pinpoint particular criteria for potential enhancement at each station, as depicted in the Sankey diagram (Figure 14). Thus, the TOD index has the potential to guide TOD policy by evaluating station zones on a regional level and pinpointing particular issues at each station on a local level. On a regional scale, it also guarantees that regional plans and policies are upheld and not overlooked, owing to domestic circumstances and preferences. Additionally, the results of the TOD index highlight the necessity for maximizing the capacity utilization of transit lines around all stations.

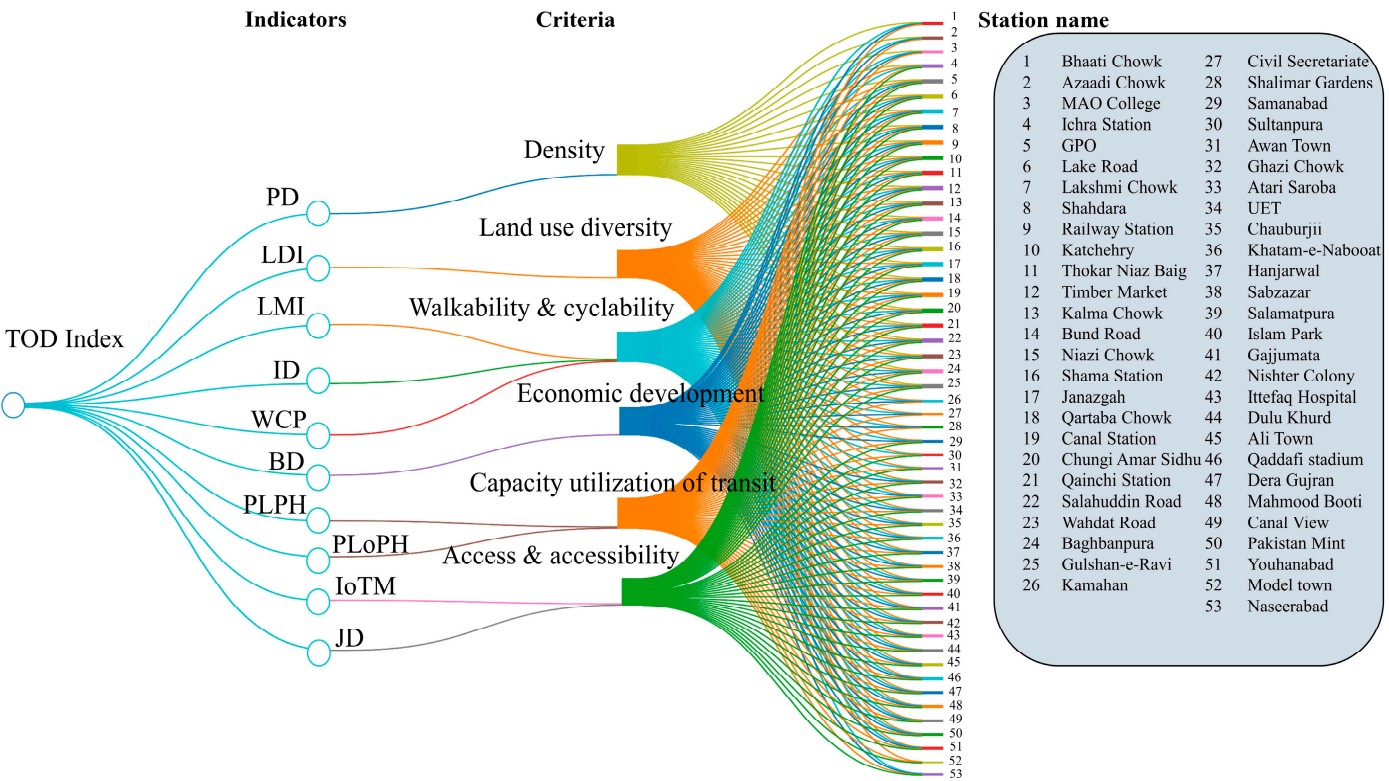

**Figure 14.** Criteria with potential for improvement at each station.

## 8. Conclusions and Policy Implications

### 8.1. Conclusions

The current case study aimed to assess the feasibility of utilizing the TOD index in a Pakistani city that is already densely populated and has mixed land use. The paper explains how the index enables a direct association of transit station areas across an urban area. We utilized six criteria to calculate our TOD index, which assesses both development and transit-related characteristics that affect travelers' choices to use public transport over cars. We conducted a sensitivity analysis to examine the impact of changes in indicator values on our results. Recommendations for improvement were derived from the TOD index findings.

The study demonstrated that the TOD index applies to Pakistani megacities. So, each station should be addressed based on its distinct characteristics and challenges to achieve optimal TOD index outcomes. The findings indicate that the highest index score of 0.61 is primarily concentrated in the inner-city area of the region. Conversely, outer areas with scores ranging from 0 to 0.32 exhibit the potential for nearing TOD status, and enhancing their scores could yield benefits for the entire district. This could further enhance the accessibility, functionality, and vibrancy of each station compared to its previous state. The TOD index results emphasize the need to optimize the capacity utilization of transit lines at all stations. The combined daily peak ridership of the Green line (180,000) and Orange line (245,000) represents 5% of the city's total daily motorized trips. A well-functioning

mass transit system typically serves at least 20% of a city's daily ridership, and as such, the current figure falls significantly below this threshold, leading to questions about its cost-effectiveness. Successful public transport projects in Tokyo, Paris, New York City, and London have achieved significant shares of daily trips, with Tokyo and Paris reaching 60% of all daily trips, New York City managing around 55% of daily trips, and London handling approximately 37% of daily trips.

The Master Plan of Lahore 2050 also mentioned that both lines are operating below their projected capacity. They suggested implementing a TOD along these corridors, with the Thokar Niaz station and its vicinity proposed as a pilot project to showcase the TOD redevelopment strategy. According to our findings, the Thokar Niaz Baig station ranks 11th with a score of 0.5, indicating potential for improvement. However, the Bhaati Chowk and Azaadi Chowk stations should be prioritized for the phased implementation of transit-oriented development (TOD) strategies. In conclusion, potential recommendations emerge from the current research considering various limitations, indicating the need for further investigations to address these constraints.

### 8.2. Policy Implications

As TOD is a planning theory, it depends on the idea of establishing context-specific resolutions to local problems. Pakistani megacities need a practical plan for TOD that is applicable and workable, depending on local circumstances, capabilities, and existing resources. Firstly, it is advised to devise solutions tailored to the context, determining appropriate priorities and levels of intervention necessary to optimize TOD potential. This may involve enhancements at the network level, improving street connectivity, or upgrading station areas. The master plan of Lahore 2050 proposes increasing allowable building heights to 8 to 12 floors within the TOD zone (500 m). A unified policy solely designating buffer areas for redevelopment and densification is inadequate. Just piling up floor space next to transit may or may not result in public transit ridership. It may create transit-adjacent development (TAD). Likewise, strategies that are appropriate for a transfer station may not be successful for other types. In addition, good intensification policies at a developed center (GPO, Lakshmi) might be contrasting from a Greenfield low-density area (Youhanabad, Naseerabad). The peripheral stations like Gajjumata (Green line) and Dera Gujran (Orange line) provide substantial Greenfield areas for development while inner city stations like Bhatti and Railway offer mostly urban infill and renewal choices. Hence, a TOD index facilitates TOD planning by enabling the comparison of stations situated across various city zones. As such, planners and designers should meticulously consider the detailed criteria and indicators before advocating for TOD implementation at a node (see Figure 14).

Secondly, the TOD index results indicate the need for capacity utilization of transit lines around all stations, although a modal shift from personal cars to public transit is the most difficult task to perform anywhere in the world. Numerous cities have applied a variety of actions to reduce the number of cars on the road, such as improved public transportation, time-restricted usage of select routes (full time or during specific hours), congestion charging, parking fees, and others. Cities like Singapore, Hong Kong, Jakarta, Delhi, Bogota, Abu Dhabi, and Cairo have demonstrated how to sustain economic growth by creating jobs, providing their citizens with high standards of living, and reducing carbon footprints. The Urban Unit [17] also recommended immediate interventions for improving air quality in Lahore, including the use of low-sulphur fuels, wall-to-wall road pavement, railway freight transport, promotion of electric vehicles, and public awareness for increased mass transit and carpooling. Integrated planning and more cost-effective resource usage can increase prosperity and social inclusion in cities at lower prices. Likewise, cities with lower-income populations, including Ahmedabad, Addis Ababa, and Dar es Salaam, have successfully adapted compact and transit-oriented expansion to their environments [76].

Additionally, we need strategies to discourage private transport by giving a better choice to travel within the city. BRT in Ottawa supports ridership by controlling the park-

and-ride facilities and encouraging the usage of feeder facilities to access transit stations, thus making more space available for development. We need area management plans because, in the South Asian context, vehicles are a status symbol and cannot go away easily. So, share those vehicles like carpooling, also known as ride-sharing, a transportation concept where multiple individuals share a single vehicle to travel together to a common destination. It can help alleviate parking challenges and reduce the overall environmental impact of transportation. Treating parking as real estate, we can design the street in a way that supports walkability as every street is a design exercise. Parking plazas should be constructed on crowded stations so that people with their vehicles can use this facility. Lahore city needs low crowding of vehicles but high crowding of people. Build and preserve open areas like amenities, green areas, playgrounds, and parks as a basic component of TODs at all levels. Moreover, the incorporation of advanced technologies in TOD plans since the start will give advantages for transit services to compete with automobiles. Smart Parking management, fare integration, and information integration are some of the basic project parts.

In light of this, placing more people near transit and mixed land uses (jobs and residential) upholds greater service and supports high ridership, empowering public transit to be more combative with the automobile by dropping trip generation and traffic congestion. Proper TODs are possible by creating a diversity of housing choices (variety of styles, categories, cost, and tenancy) within a 10-min walkable/cycling distance from a transit station. Public transit needs ridership so we need to bring riders closer to transit. Through our planning, we can make it possible for people to stay close to transit who want to use it. Affordable housing programs and transit initiatives should engage in dialogue and be incentivized to collaborate effectively. So, mixedness index results can be utilized to identify stations that need more residential mix to make balance. Furthermore, Gentrification is a critical impact of TOD as land and property prices rise, resulting in the displacement of the urban poor. So, the Government needs to work on more affordable housing complexes in those areas, by keeping gentrification in mind.

Thirdly, Pakistan has various informal travel modes (paratransit services), including Qingqi rickshaws and auto rickshaws, that are not working in developed states. Such types are flexibly developed by themselves to fulfill local requirements and are highly admired by the public because of their availability, service types, and fares. Researches also reveal that commuters in developing states are customarily inclined to use these informal modes for short distances, which can be traveled by walking or cycling [77]; featuring that careful planning is required to include these informal modes to get benefits from transit investments. Their role for access or egress trips and feeder bus routes will help identify influence zones of a particular transit station and will help to develop TOD. It can be more effective if coincides with broad cycle and bikeways with parking facilities and bicycle-friendly strategies [78,79] like in Paris, Amsterdam, Barcelona, Copenhagen, Groningen, and Stockholm.

Ultimately, while a fundamental rule for TOD is to plan for the long-term spanning a horizon of 10–20 years, interventions at enhancing accessibility to transit stops and stations provide quick wins (time frame of 2–3 years). These interventions are recommended in the Master Plan for Lahore 2050 as incremental additions for creating complete TOD projects. Some of these immediate projects may include the following:

- Multi-modal integration for different modes comprising direct access among feeder buses, auto rickshaws, Qingqi, and transit stations;
- Improved pedestrian infrastructure encompassing footpaths, street furniture, specified waiting for spaces;
- Enhanced station facilities along with commuter services;
- Integrating bicycle lanes and supporting rental or sharing schemes near transit stations;
- Park-and-ride services at deliberate transit settings;
- Recognize resource person's team for handling TOD plans.

The research findings will provide valuable support for the Lahore Master Plan 2050 and will be beneficial for future urban development initiatives in Lahore, as well as in other cities in Pakistan and developing countries aiming to implement mass transit systems.

**Author Contributions:** Conceptualization, A.A. and H.L.; methodology, A.A. and H.L.; software, A.A., H.L., H.A. and A.H.; validation, AA., H.L., H.A. and A.H.; formal analysis, A.A.; investigation, A.A.; resources, A.A., H.L., H.A. and A.H.; data curation, A.A.; writing—original draft preparation, A.A.; writing—review and editing, A.A. and H.L.; visualization, A.A., H.A. and A.H.; supervision, H.L.; project administration, A.A. and H.L. All authors have read and agreed to the published version of the manuscript.

**Funding:** This research received no external funding.

**Institutional Review Board Statement:** Not applicable.

**Informed Consent Statement:** Not applicable.

**Data Availability Statement:** Researchers interested in accessing the data for validation or further analysis can contact the corresponding author to discuss data availability and permissions.

**Acknowledgments:** We would also like to thank all authors for their advice and assistance, which kept our progress on schedule.

**Conflicts of Interest:** The authors declare no conflicts of interest.

## Appendix A

**Note:** The web diagrams are organized according to the TOD Index score ranking, with Green line stations represented in green and Orange line stations in orange, facilitating improved comprehension.

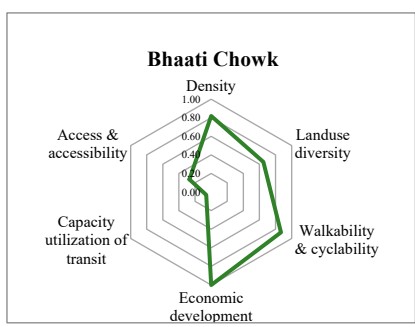
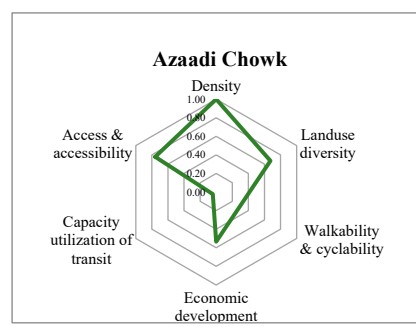
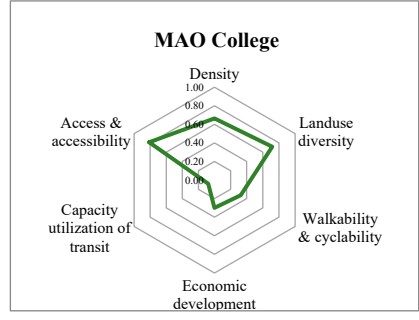
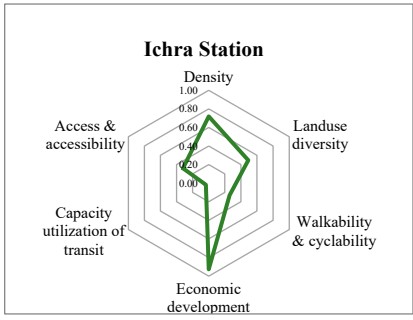
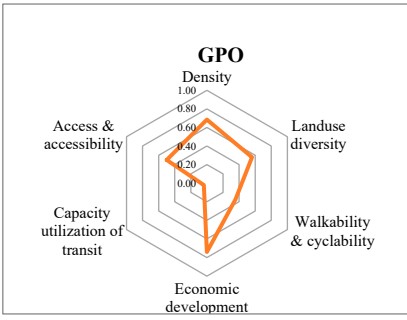
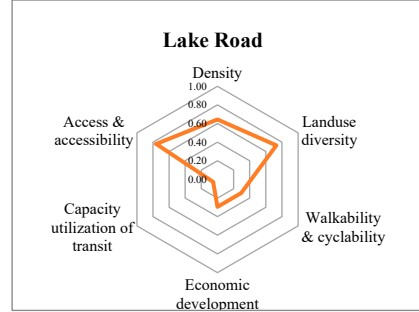

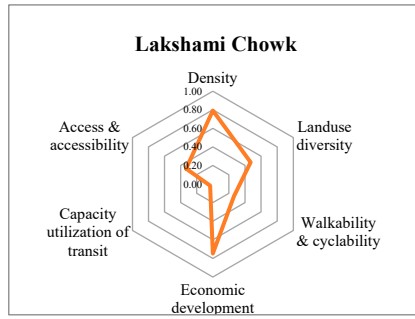

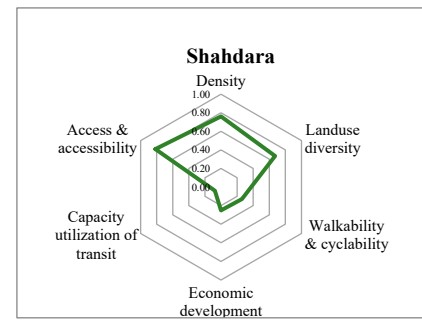

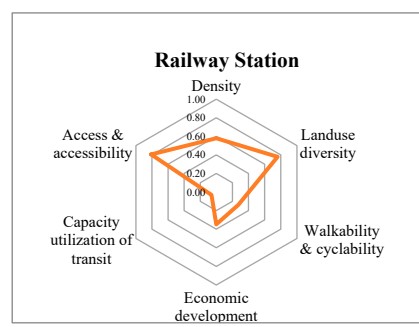

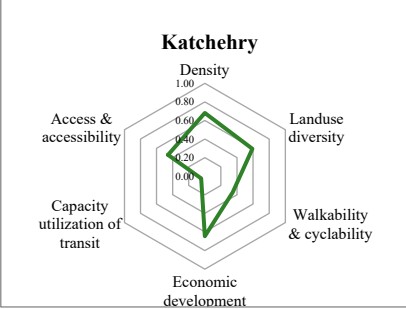

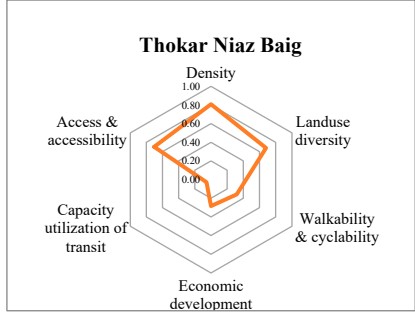

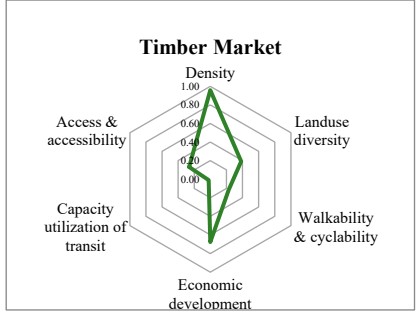

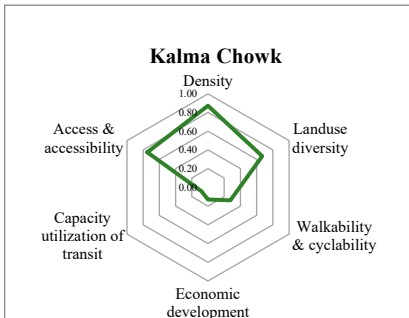

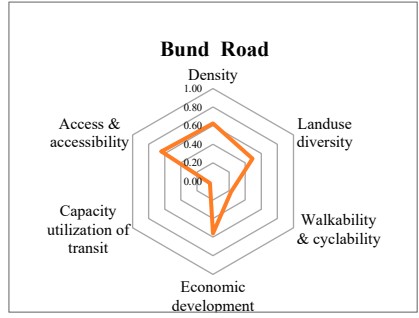

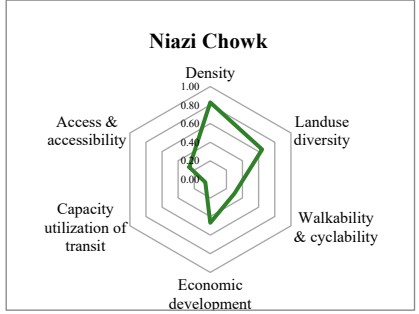

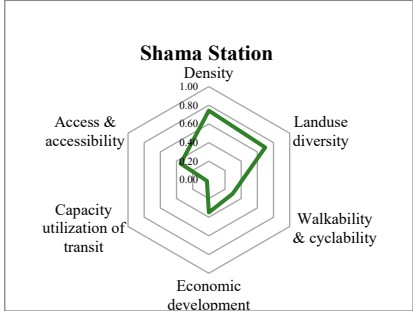

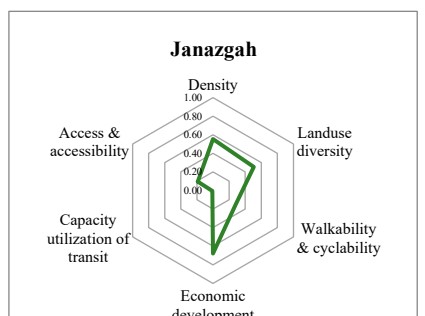

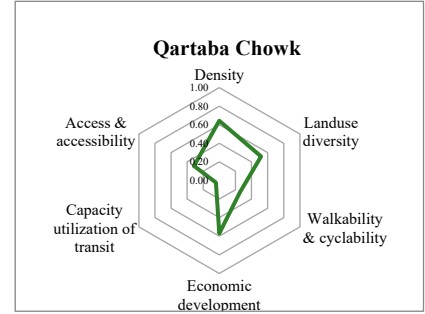

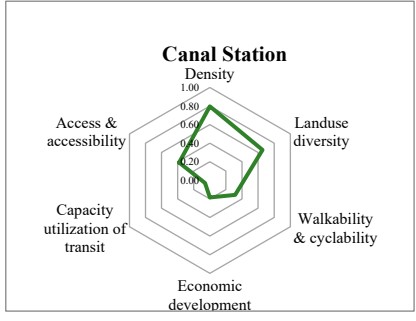

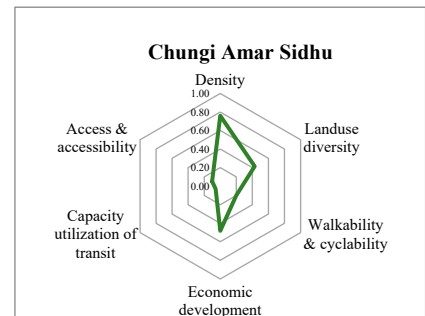

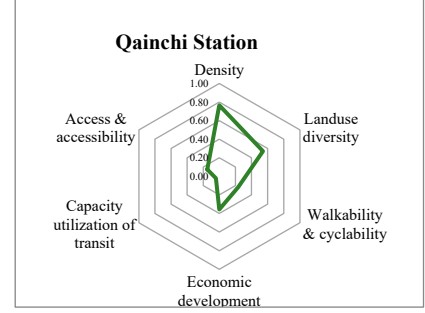

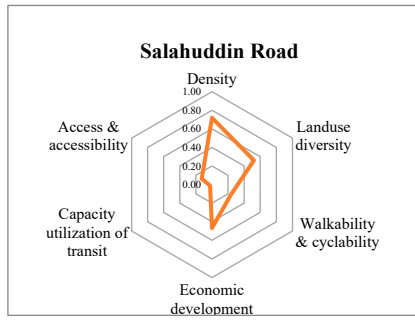

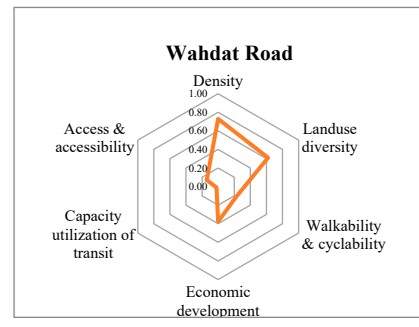

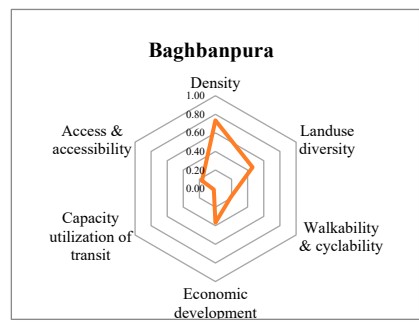

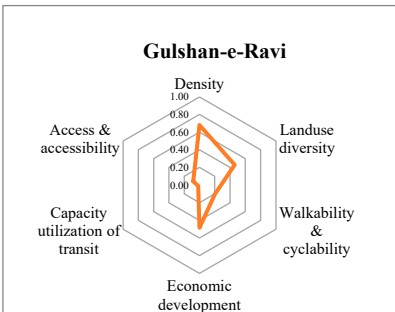

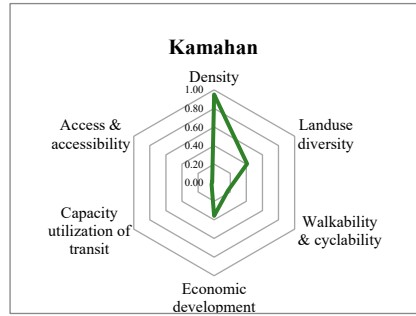

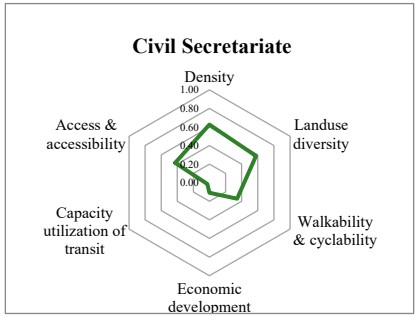

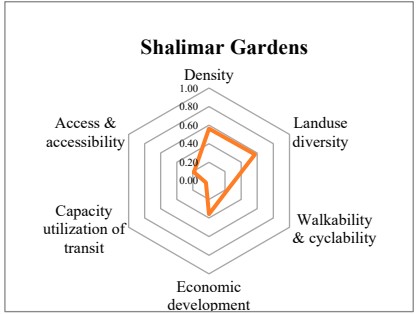

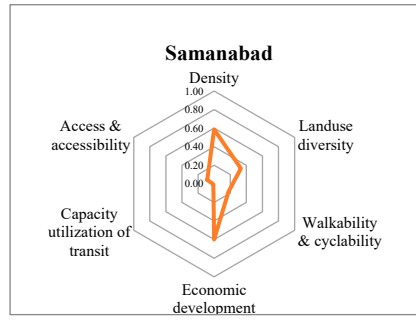

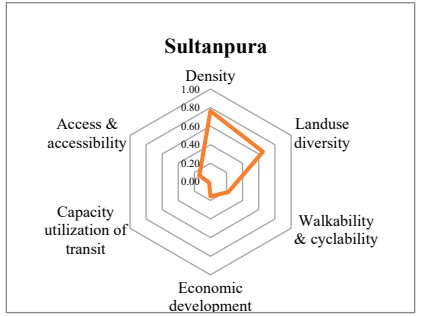

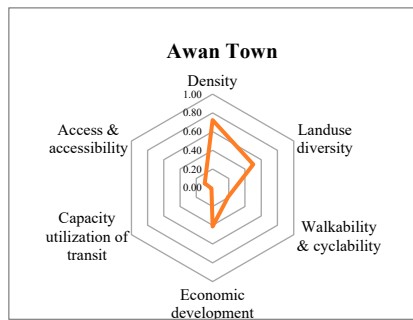

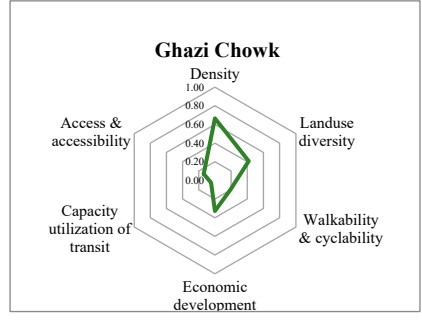

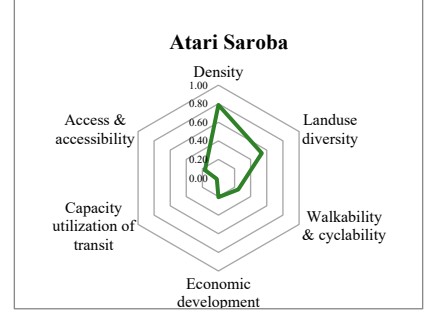

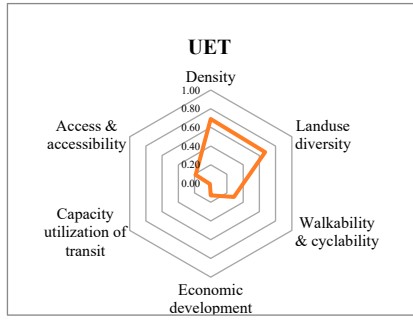

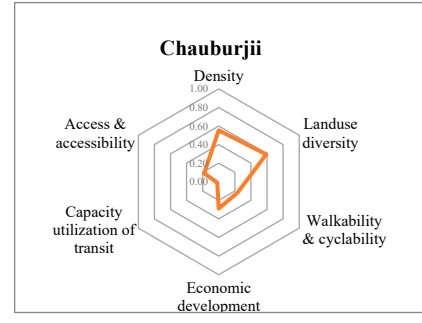

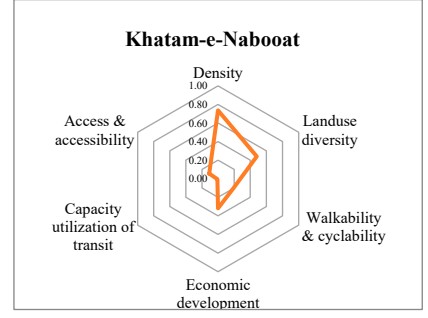

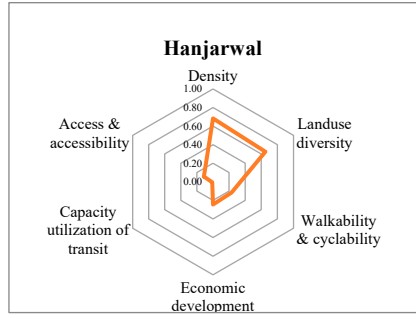

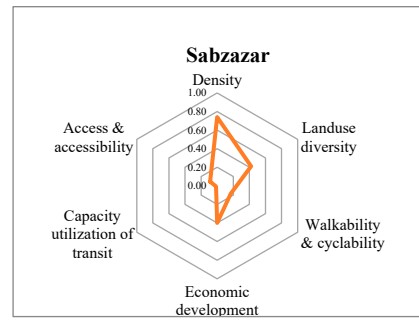

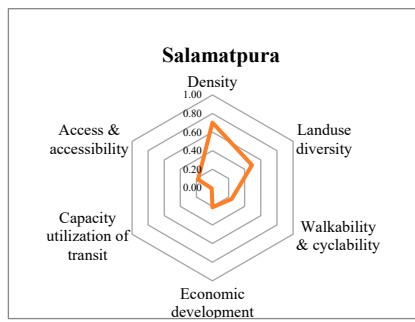

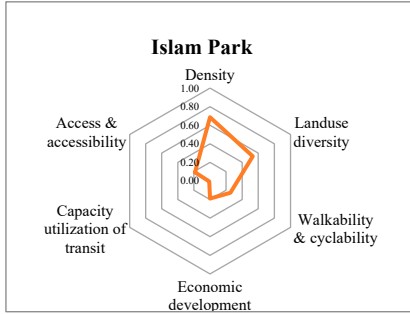

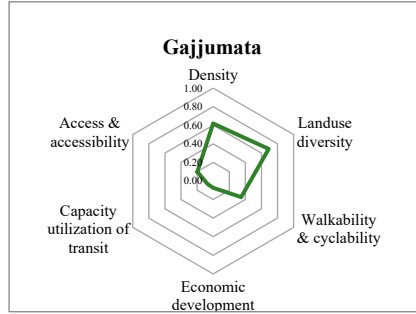

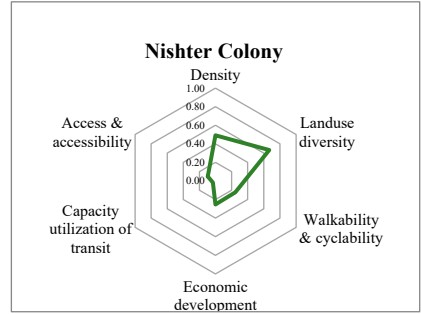

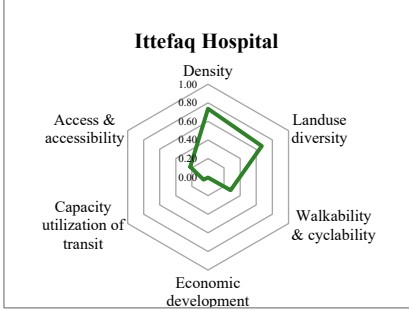

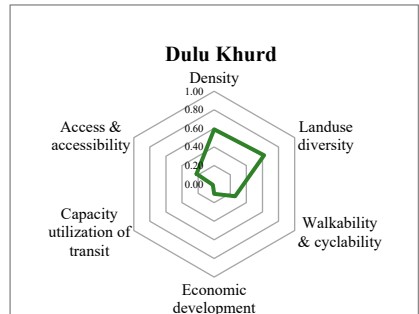

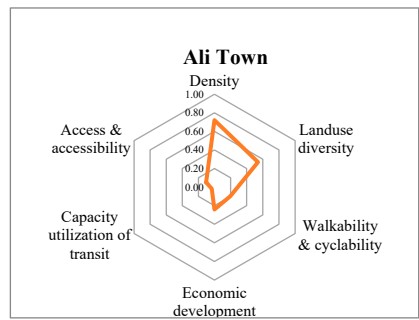

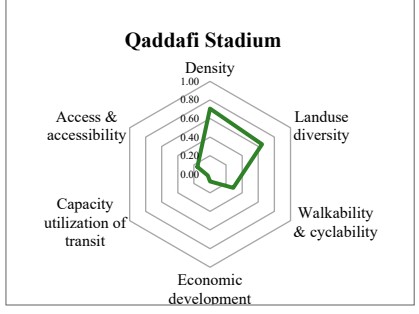

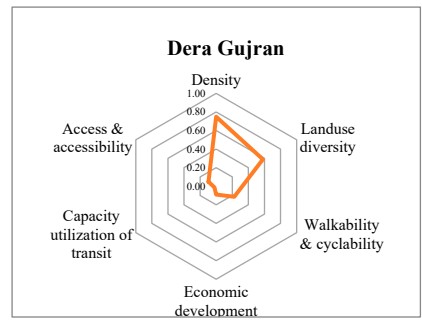

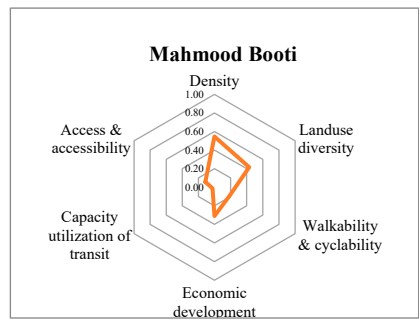

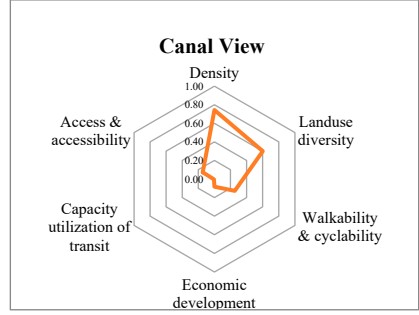
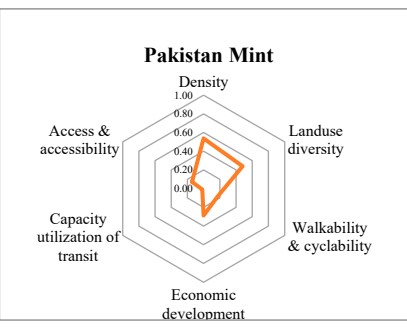
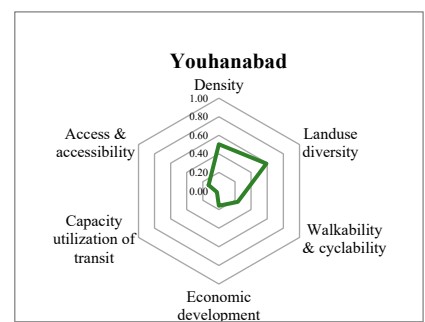
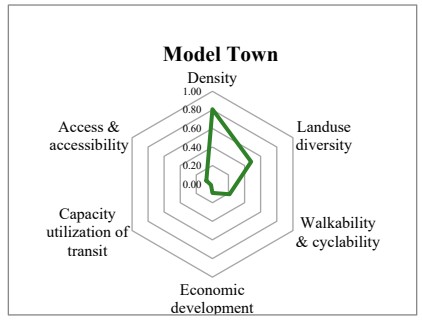
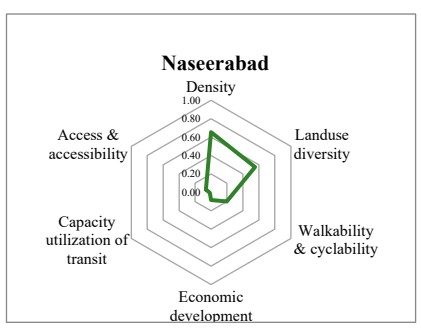

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
