# Peer review of "Measuring the Transit-Oriented Development (TOD) Levels of Pakistani Megacities for TOD Application: A Case Study of Lahore"

_sustainability, doi:10.3390/su16052209_

Round 1
Reviewer 1 Report
Comments and Suggestions for Authors
There are some observations of the manuscript, focused on the form and content. In any case, it stands out that in general it is a good work.
1) A brief summary outlining the aim of the paper, its main contributions and strengths
In general, it is a well-developed work. The strong point of this work is its structure and clarity for the coherence between the objective and results that is clear. The aim of the paper is to assess the feasibility of utilizing the Transit-Oriented Development (TOD) index in a Pakistani city (Lahore). Likewise, the following research problem is proposed: there is little prior experience and evidence to support the applicability of TOD for developing cities who need context-specific strategic action plans. That is why this study attempts to fill the gap through particular studies about TOD application in Pakistan, a developing country in South Asia.
Also, it proposes three specific objectives: a) to applying Spatial and non-spatial analytical methods (ArcGIS and SMCA) to get TOD index based on set of indicators; b) to apply spatial statistical methods (sensitivity analysis) to analyze the obtained values of TOD index; c) to guide decision makers to recognize priorities for potential use.
The article mainly contributes with to provide valuable support for the Lahore Master Plan 2050 and will be beneficial for future urban development initiatives in Lahore, as well as in other cities in Pakistan and developing countries aiming to implement mass transit systems.
2) General concept comments of the article: highlighting areas of weakness, the testability of the hypothesis, methodological inaccuracies, missing controls, etc.
In this article I did not find areas for improvement, I think it is a very rigorous investigation and very well carried out.
It should be noted that this article does not present a hypothesis, nor a research question, but it is not considered something negative, it is only verified.
3) Specific comments referring to line numbers, tables or figures that point out inaccuracies within the text or sentences that are unclear
The article has some elements to improve throughout the manuscript that are indicated in the following points:
- Figure 10: the legend is illegible.
- citations: several citations are not with the Vancouver style (introduction, literature review).
Author Response
|
Thank you very much for taking the time to review this manuscript. Please find the detailed responses below and the corresponding corrections highlighted in the re-submitted file. |
|
Point-by-point response to Comments |
|
Comments 1: Figure 10: the legend is illegible |
|
Response 1: Thank you for pointing this out. We agree with this comment. Therefore, we have improved the legend to be legible. See figure 12. At line 596 In the revised manuscript |
|
Comments 2: - citations: several citations are not with the Vancouver style (introduction, literature review). |
|
Response 2: Agree. We have, accordingly, modified the citations to match the Vancouver style. See section 1 & 2 in the resubmitted file |
Reviewer 2 Report
Comments and Suggestions for Authors
Dear authors,
Please, enter the following technical changes:
- Line 98: instead ''proposed by (9,11)'' enter the author's name ''propsed by Evans et al., 2007 and Singh et al., 2017''
- Text in the Figures 3,4,5,6 and 10 is not legible. Please. make it visible.
Author Response
|
Thank you very much for taking the time to review this manuscript. Please find the detailed responses below and the corresponding corrections highlighted in the re-submitted file. |
|
Point-by-point response to Comments |
|
Comments 1: - Line 98: instead ''proposed by (9,11)'' enter the author's name ''proposed by Evans et al., 2007 and Singh et al., 2017'' |
|
Response 1: Thank you for bringing this to our attention. We concur with this observation. Consequently, we have implemented the suggested changes. Please refer to line 83 in the revised manuscript. |
|
Comments 2: -Text in the Figures 3,4,5,6 and 10 is not legible. Please. make it visible. |
|
Response 2: Agree. We have, accordingly, improved the images to be legible. See figure 3 at line 186, figure 4 at line 368, figure5 at line 370, figure 6 at line 399, figure8 at line 476 and figure 12 at line 595. In the revised manuscript |
Reviewer 3 Report
Comments and Suggestions for Authors
The idea seems good, but its representation and writing are not catchy. The authors should address the below concerns before going for any action:
1) The title should be revised.
2) Why TOD is described many times? Its full form at first occurrence is enough.
3) In the Introduction, the authors mentioned that many had applied the TOD concept. Right? It will be better to highlight the core difference. The author should target ONE article where they found potential limitations.
4) Figures 1 and 2 are not visible. Enhance its quality.
5) The authors should add a comparison table in the literature review section.
6) The structure of the paper needs to be revised. Shifting the Research design section after the literature review will be better.
7) Many figures showing the statistical results are added before the results section. Why?
Comments on the Quality of English LanguageThe paper needs to be proofread a couple of times.
Author Response
|
Thank you very much for your thorough review of the manuscript, which has undoubtedly enhanced the clarity and impact of our work. Below, you will find our detailed responses to your comments and the corresponding revisions highlighted in the resubmitted file. |
|||
|
Point-by-point response to Comments |
|||
|
Comments 1: The title should be revised. |
|||
|
Response 1: Thank you for notifying us of this matter. We agree with your observation and have accordingly modified the title to enhance clarity. Please refer to the revised manuscript for the updated title. |
|||
|
Comments 2: - Why TOD is described many times? Its full form at first occurrence is enough. |
|||
|
Response 2: Thank you for bringing this technical issue to our attention. We have addressed it by revising the manuscript accordingly. |
|||
|
Comments3: In the Introduction, the authors mentioned that many had applied the TOD concept. Right? It will be better to highlight the core difference. The author should target ONE article where they found potential limitations. |
|||
|
Response 3: Thank you for your notification. We have revised the Introduction section accordingly. Please refer to this section for an overview of the changes, and specifically lines 89, 90, and 91 for the detailed updates. |
|||
|
Comments 4: Figures 1 and 2 are not visible. Enhance its quality. |
|||
|
Response 4: Agree. We have, accordingly, improved the images to be legible. See figure 1 at line 79 and figure 2 at line 81 in the revised manuscript |
|||
|
Comments 5: The authors should add a comparison table in the literature review section. |
|||
|
Response 5: We agree with your observation and have taken action accordingly. A comparison table has been added to the literature review section. Please refer to Table 1 at line 154 in the revised manuscript for further details. |
|||
|
Comments 6: The structure of the paper needs to be revised. Shifting the Research design section after the literature review will be better. |
|||
|
Response 6: We sincerely appreciate your thoughtful critique, which has been instrumental in refining our ideas and improving our presentation. Notably, we have restructured the manuscript, with the research design section now following the literature review. Please refer to the revised manuscript for further details in this section from line 155. |
|||
|
Comments 7: Many figures showing the statistical results are added before the results section. Why? |
|||
|
Response 7: Thank you for bringing this matter to our attention. In the introduction, at lines 106 and 107, we mentioned that sections 5, 6, and 7 detail the TOD index results and analysis. However, we acknowledge that the heading of section five may cause confusion. Therefore, we have revised it for better clarity. Please refer to line 414 for the revised heading in the manuscript. |
|||
|
Reviewer 4 Report
Comments and Suggestions for Authors
· Originality/Novelty
The paper provides an extensive literature review on TOD concept and criteria used for TOD station evaluation index. The paper novelty is not very high, but the application of previous methods specifically for the two transit lines in Lahore megacity is original and of interest. The interest comes especially from detailed information on the very densely population megacity Lahore.
However, the title may produce some confusion because it announces the readiness evaluation (which is an index itself) but in the paper there is no correlation/discussion between TOD index and readiness level (or however this is defined).
- Therefore, authors are advised to simplify the title and change it to match the content, so as not to confuse the reader.
· Significance
The paper has significance especially for the decision-makers in Lagore: the two lines of transit were built on the top-to-down decision and now, further actions are needed for making them effective and efficient with a high utilization.
· Quality of Presentation
Section 3 is the best quality part of the paper. However,
- The introduction and literature review are unnecessarily extensive; they should be more concise.
- Some statements need references (e.g. line 337 and 364-366).
- For better highlighting some figures (e.g. Figures 7, 8 and 9) are recommended to be transposed in suggestive representation on maps: it is difficult to follow each station to make comparisons/analyses.
- Section 7- Interference and discussion requires a much more narrowed and synthesized representation for both Figure 11 and Table 6.
- The last section 8. Conclusions and policy implications should be divided into two parts, to clearly highlight the conclusions of the paper. In this way, the reader could keep in mind the conclusions and contributions of the paper.
· Scientific Soundness
There are two issues that require a clearer explanation as follows:
- In sub-section 4.3.2 Weight of indicators, the authors consider equal weights for all criteria. The weighting has an important impact on the entire model and the results of its application, and therefore requires clear explanations, justifications and substantiations. It is also then necessary to demonstrate that this simplistic assumption of equal weights does not distort the results.
- In Section 5.3 Mixture, equation (3) is not described clearly enough, because MI can have the value 0 when no non-residential activity is located in area i (i.e. Sc=0), and it has the value 1 if which area is exclusively residential, i.e. Sr=0. In lines 486-487, the explanations are contradictory and cause confusion.
· Interest to the Readers
In the opinion of the reviewer, many readers could be interested in the topic of the paper, in particular, decision-makers in strategic planning.
However, the paper may disappoint some readers due to the lack of clarity regarding the scientific method used, discussions and conclusions.
· Overall Merit:
The article has the merit of using large and different spatial data (land use, transportation, and demographics) in ArcGIS.
· English Level:
The English language is sufficiently appropriate and understandable.
Author Response
|
Thank you very much for your thorough review of the manuscript, which has undoubtedly enhanced the clarity and impact of our work. Below, you will find our detailed responses to your comments and the corresponding revisions highlighted in the resubmitted file. |
|
Point-by-point response to Comments |
|
Comments 1: The title may produce some confusion because it announces the readiness evaluation (which is an index itself) but in the paper there is no correlation/discussion between TOD index and readiness level (or however this is defined). Therefore, authors are advised to simplify the title and change it to match the content, so as not to confuse the reader. |
|
Response 1: Thank you for notifying us of this matter. We agree with your observation and have accordingly modified the title to enhance clarity. Please refer to the revised manuscript for the updated title. |
|
Comments 2: The introduction and literature review are unnecessarily extensive; they should be more concise. |
|
Response 2: Thank you for bringing this to our attention. We have therefore revised the Introduction and Literature Review sections. The Introduction now concludes at line 108, compared to its previous ending at line 124. Similarly, the Literature Review now ends at line 154, whereas it previously concluded at line 196. |
|
Comments3: Some statements need references (e.g. line 337 and 364-366). |
|
Response 3: Thank you for mentioning. We have added references to the specified lines accordingly. Please refer to line 181 and lines 208-210 in the revised manuscript for the added references. |
|
Comments 4: For better highlighting some figures (e.g. Figures 7, 8 and 9) are recommended to be transposed in suggestive representation on maps: it is difficult to follow each station to make comparisons/analyses. |
|
Response 4: Thank you for raising this valid point. We have indeed transposed these figures according to their ranking to facilitate better understanding and comparison. Please refer to Figures 9, 10, and 11 in the revised manuscript for the updated arrangements. |
|
Comments 5: Section 7- Interference and discussion requires a much more narrowed and synthesized representation for both Figure 11 and Table 6. |
|
Response 5: Thank you for bringing this weakness to our attention. We have addressed it by synthesizing our inferences and discussion into web diagrams for the 4 top-scoring and 4 lowest-scoring stations, as depicted in Figure 13. Additionally, potential improvements for each station are presented in the form of a Sankey diagram in Figure 14 in the revised manuscript. |
|
Comments 6: The last section 8. Conclusions and policy implications should be divided into two parts, to clearly highlight the conclusions of the paper. In this way, the reader could keep in mind the conclusions and contributions of the paper. |
|
Response 6: Thank you for this insightful suggestion. We have divided the Conclusions and policy implications into two separate parts for improved clarity and comprehension. Please refer to sections 8.1 and 8.2 in the revised manuscript for the updated content. |
|
Comments 7: In sub-section 4.3.2 Weight of indicators, the authors consider equal weights for all criteria. The weighting has an important impact on the entire model and the results of its application, and therefore requires clear explanations, justifications and substantiations. It is also then necessary to demonstrate that this simplistic assumption of equal weights does not distort the results. |
|
Response 7: Indeed, your point regarding the importance of criteria weighting is valid. We have accordingly included further explanation in sub-section 4.3.2 to enhance clarity and justify the selection of weights. |
|
Comments 8: In Section 5.3 Mixture, equation (3) is not described clearly enough, because MI can have the value 0 when no non-residential activity is located in area i (i.e. Sc=0), and it has the value 1 if which area is exclusively residential, i.e. Sr=0. In lines 486-487, the explanations are contradictory and cause confusion. |
|
Response 8: Thank you for bringing this critical mistake to our attention. We have revised the explanation of MI to eliminate any confusion. Please refer to lines 463-465 in section 5.3 of the revised manuscript for the updated explanation. |
Round 2
Reviewer 3 Report
Comments and Suggestions for Authors
The reviewer's comments are addressed properly. Accepted for publication
Reviewer 4 Report
Comments and Suggestions for Authors
The authors tried to improve the paper and correct the items discussed in the first round of revision. In this way the article is publishable.